# Identification of SARS-CoV-2 inhibitors targeting Mpro and PLpro using in-cell-protease assay

Anoop Narayanan[1], Manju Narwal[1], Sydney A. Majowicz [2], Carmine Varricchio[3], Shay A. Toner[1], Carlo Ballatore[4], Andrea Brancale [3], Katsuhiko S. Murakami [1] & Joyce Jose [1,2✉]

SARS-CoV-2 proteases Mpro and PLpro are promising targets for antiviral drug development. In this study, we present an antiviral screening strategy involving a novel in-cell protease assay, antiviral and biochemical activity assessments, as well as structural determinations for rapid identification of protease inhibitors with low cytotoxicity. We identified eight compounds with anti-SARS-CoV-2 activity from a library of 64 repurposed drugs and modeled at protease active sites by in silico docking. We demonstrate that Sitagliptin and Daclatasvir inhibit PLpro, and MG-101, Lycorine HCl, and Nelfinavir mesylate inhibit Mpro of SARS-CoV-2. The X-ray crystal structure of Mpro in complex with MG-101 shows a covalent bond formation between the inhibitor and the active site Cys145 residue indicating its mechanism of inhibition is by blocking the substrate binding at the active site. Thus, we provide methods for rapid and effective screening and development of inhibitors for blocking virus polyprotein processing as SARS-CoV-2 antivirals. Additionally, we show that the combined inhibition of Mpro and PLpro is more effective in inhibiting SARS-CoV-2 and the delta variant.

[1] Department of Biochemistry and Molecular Biology, The Pennsylvania State University, University Park, PA, USA. [2] Huck Institutes of the Life Sciences, The Pennsylvania State University, University Park, PA, USA. [3] School of Pharmacy and Pharmaceutical Sciences, Cardiff University, King Edward VII Avenue, CF10 3NB Cardiff, UK. [4] Skaggs School of Pharmacy and Pharmaceutical Sciences, University of California, San Diego, 9500 Gilman Drive, La Jolla, CA 92093, USA. ✉email: jxj321@psu.edu

The coronavirus disease 2019 (COVID-19) pandemic caused by the severe acute respiratory syndrome-coronavirus-2 (SARS-CoV-2) has infected over 293 million people worldwide and has caused nearly 5.4 million deaths across the world since it was first isolated in December 2019 from Wuhan, China[1]. The COVID-19 pandemic has thus far killed 820,000 people in the USA alone (CDC COVID Data Tracker - gov.cdc.covid—CDC https://covid.cdc.gov). The main clinical characteristics of COVID-19 include fever, cough, and shortness of breath that can progress rapidly to respiratory and cardiac failure requiring mechanical ventilation[2]. The elderly, immuno-compromised, and those with co-morbid metabolic, pulmonary, and cardiac conditions are at greater risk of death from COVID-19[1]. SARS-CoV-2 is a positive-sense single-stranded RNA virus closely related to SARS-CoV and Middle East respiratory syndrome coronavirus (MERS-CoV) that belong to the genus beta-coronavirus in the *Coronaviridae* family[3]. The emergence of SARS-CoV-2 has created an urgent need to develop antiviral agents and vaccines[4]. Even though effective vaccines have been developed against COVID-19, limited progress has been made in developing antivirals to treat COVID-19[5]. The COVID-19 vaccine inequity, vaccine hesitancy, and the appearance of SARS-CoV-2 variants of concern (VOC) with increased ability to spread and potential to escape from both vaccine and natural infection immunity highlight the importance of developing antiviral drugs to combat SARS-CoV-2 infections[6]. SARS-CoV-2 variants such as B.1.1.7, first identified in the United Kingdom, and B.1.617, first identified in India, have been linked to increased transmissibility and an ability to evade immune protection. B.1.617 has since been reported in 20 other countries[7]. On 24 November 2021, a new SARS-CoV-2 variant of concern B.1.1.529 with a large number of mutations was reported to WHO (www.who.int) from South Africa, and the VOC was named Omicron. In this context, the repurposing of existing drugs may provide opportunities for relatively rapid identification of clinical candidates[8].

The SARS-CoV-2 has a 29.9 kb single-stranded, non-segmented RNA genome with a 5′ Cap and a 3′ poly-A tail. The 5′ 20 kb of the genome codes for 2 large open reading frames (ORF 1a/b) producing two polyproteins pp1a and pp1ab[4–6]. The polyproteins are cleaved by the viral papain-like protease (PLpro) and the 3C-like cysteine protease (3 CLpro) also known as Main protease (Mpro) to produce the non-structural proteins (nsp) 1–16. PLpro cleaves at its *LXGG* recognition sites at nsp1, nsp2, and nsp3 while Mpro cleaves the remaining downstream non-structural proteins (nsp4-16)[9,10]. Even though all viral enzymes that participate in coronavirus replication are potentially druggable targets, antiviral studies with small-molecule inhibitors have been focused on the RNA-dependent RNA polymerase (RdRP) and the two viral proteases PLpro and Mpro[9,11,12]. High-throughput assays have been effectively used for large-scale screening of existing drugs to identify potential antiviral leads for SARS-CoV-2. Carmofur and Ebselen that inhibit SARS-CoV-2 infection of Vero cells were identified from a virtual structure-based and high-throughput screening of a library of about 10,000 compounds[13]. Similarly, Apilimod, MDL-28170, and ONO 5334 that inhibit SARS-CoV-2 were identified by profiling a library of 12,000 clinical-stage or Food and Drug Administration (FDA)-approved small molecules[14]. Lopinavir–ritonavir is a drug combination used to prevent and treat HIV infection and works by inhibiting protease activity[15]. Lopinavir showed in vitro activity against SARS-CoV and has been effective in improving the clinical outcome of MERS in nonhuman primates[16]. In addition, viral and host-factor-targeting agents, combined with drugs that directly target viral enzymes, could lead to a therapeutic regimen to treat COVID-19[17]. Camostat mesylate, which inhibits the plasma membrane-associated host serine protease, TMPRSS2, has

been shown to block the SARS-CoV cell entry mechanism[18]. Antivirals such as Remdesivir, Favipiravir, and Galidesivir, targeting RdRP, have shown inhibitory activities against SARS-CoV-2[19–22]. Remdesivir was granted emergency use authorization for SARS-CoV-2 from the U.S. FDA on 1 May 2020[23]. Although Remdesivir can shorten infection times and may have clinical benefits in patients with severe COVID-19, it did not significantly improve survival[24,25]. An oral RdRP inhibitor, Molnupiravir (MK-4482, EIDD-2801) was found effective in patients early in the course of their illness[26]. The FDA and United Kingdom Medicines and Healthcare products Regulatory Agency (MHRA) has granted authorization for Molnupiravir to treat mild-to-moderate COVID-19.

Since there are no FDA-approved antiviral strategies involving protease inhibitors to treat COVID-19, considerable efforts are being made to find compounds that inhibit Mpro and PLpro with desirable pharmacokinetic properties. The active site of SARS-CoV-2 Mpro is comprised of the catalytic dyad Cys145 and His41, which is part of a chymotrypsin-like fold resembling the picornavirus 3C proteinases[27,28]. The bisulfate-based prodrug, GC376, and the corresponding active aldehyde, GC373 which inhibit Feline Coronavirus Mpro have been shown to inhibit SARS-CoV-2 replication in cell culture[29]. The clinically approved hepatitis C virus (HCV) drug Boceprevir has been shown to inhibit SARS-CoV-2. The structure of SARS-CoV-2 Mpro in complex with Boceprevir has shown the mechanism of inhibition[8]. An antineoplastic drug, Carmofur, has been shown to inhibit the SARS-CoV-2 replication in cells by covalently modifying the catalytic Cys145 of Mpro[13,30]. Calpain inhibitors II and XII that are active against host protease cathepsin L, which is critical for SARS-CoV-2 entry, have also been shown to inhibit the protease activity of Mpro[31]. An oral antiviral compound PF-07321332 from Pfizer, specifically designed to inhibit SARS-CoV-2 Mpro modifies the active site Cys145 with its nitrile warhead, is considered a good candidate antiviral and is currently undergoing trials (NCT04756531, NCT04909853, NCT05011513, ClinicalTrials.gov). The oral antiviral PAXLOVID™, which is a combination of PF-07321332, and HIV drug ritonavir that slows down the breakdown of PF-07321332, was found to reduce the risk of hospitalization or death by 89% compared to placebo in non-hospitalized high-risk adults with COVID-19[32–34]. In December 2021, the FDA approved the emergency use authorization of Pfizer's Paxlovid to treat mild-to-moderate COVID-19 in adults and pediatric patients 12 years of age and older (www.fda.gov).

PLpro is a cysteine protease, part of the nsp3 multi-domain protein, and cleaves in trans between nsp1/2, nsp2/3, and nsp3/4[35]. Additionally, PLpro of MERS-CoV and SARS-CoV have been shown to contribute to viral pathogenesis by modifying the host innate immune response to viral infection[36]. The structure of PLpro has similarities to cellular deubiquitinases (DUBs) such as human USP12 and USP14[9,37]. Therefore, peptidomimetic inhibitors might interfere with DUBs leading to side effects, and classes of non-peptidic, naphthalene-based reversible drugs like GRL0617 are preferred for PLpro inhibition[38,39]. The SARS-CoV replication inhibitor GRL0617 is a PLpro inhibitor identified from a library of compounds using a fluorescence-based high-throughput screen[37]. The mechanism of inhibition was elucidated from an X-ray structure of PLpro in complex with GRL0617, which showed the catalysis at the active site is shut down by a loop closure[37]. Previously developed SARS PLpro inhibitors, including Rac5c, have also demonstrated antiviral efficacy against SARS-CoV-2 in cell culture[40].

New SARS-CoV-2 variants frequently emerging with increased ability to spread, spillover, or escape from immunity underscores the requirement of new and improved antiviral therapies[6,41,42]. Furthermore, the selective pressure from S-specific antibodies induced from a vaccine or original SARS-CoV-2 infection could

promote the acquisition of additional mutations to cause a change in viral antigenicity that would allow a SARS-CoV-2 variant to escape from immune responses[41]. Several SARS-CoV-2 variants with mutations in the nonstructural and structural proteins with increased viral transmission and potential to escape from the vaccine and natural infection immunity are causing concerns worldwide[43]. The B.1.351 variant, originally identified in South Africa, includes several mutations within the structural and nonstructural proteins raising significant concerns about alterations to viral fitness, transmission, and disease[44]. SARS-CoV-2 variants B.1.1.7 (alpha) and B.1.351[45] are reported to have increased transmission and resistance to antibody neutralization[46]. The SARS-CoV-2 Omicron variant B.1.1.529 harbors more than 30 changes to the spike protein and has an increased risk of reinfection compared to other VOCs. Omicron appears to be rapidly spreading across South Africa and other countries (www.who.int). There remains an urgent need for more SARS-CoV-2 therapeutic agents targeting proteins other than S protein to combat the COVID-19 pandemic effectively.

The development of broad-spectrum antiviral drugs against a wide range of coronaviruses is the ultimate treatment strategy for circulating and emerging coronavirus infections[47]. Here we show that repurposing the FDA-approved pharmaceutical drugs targeting Mpro and PLpro is an effective strategy for identifying antivirals against SARS-CoV-2 and its Delta variant. We have identified eight SARS-CoV-2 inhibitors using a novel in-cell protease assay (ICP) that measures the protease activities of Mpro and PLpro based on the subcellular localization of a cleaved fluorescent protein in live cells. The selected compounds showed dose-dependent antiviral activities against SARS-CoV-2 in Huh-7.5 cells with $EC_{50}$ values less than 1 μM. Furthermore, we show that treating cells with a combination of Mpro and PLpro inhibitors had an additive antiviral effect inhibiting the replication of SARS-CoV-2 Delta variant with no significant cytotoxicity. Docking studies with compounds that are effective against PLpro revealed that compound binding is likely to prevent substrate access to the active site. Finally, using X-ray crystallography, we show that MG-101 (Calpain inhibitor I), forms a covalent bond with the active site cysteine residue of Mpro for blocking the substrate binding at the active site. Taken together, our results clearly indicate that inhibitors of Mpro and PLpro significantly reduce SARS-CoV-2, and can be used in combination for an enhanced antiviral effect. Protease inhibitors hold considerable promise as candidate therapeutics against SARS-CoV-2 and for pandemic preparedness in the event of emerging coronavirus variants similar to MERS and SARS-CoV with a higher mortality rate.

## Results

### Development of an ICP assay for screening inhibitors of SARS-CoV-2 Mpro and PLpro.

We designed and standardized ICP assays to identify inhibitors of SARS-CoV-2 proteases Mpro and PLpro in human cell lines (Fig. 1). Although several assays are available, including fluorescence-, fluorescence resonance energy transfer (FRET)-, and luciferase-based assays for screening protease inhibitors, we developed this assay to measure the SARS-CoV-2 protease inhibition specifically in live cells, which can also directly measure the cytotoxicity of compounds. The Mpro ICP assay construct encodes a fluorescent protein, mEmerald with C-terminal nuclear localization signal (NLS) of the trafficking of the cleaved mEmerald to the nucleus, and an NS2B protein sequence from Zika virus (ZIKV) linked to the Mpro coding region (amino acids 3258 to 3575 of SARS CoV-2 pp1a) for anchoring of the protease to the ER membrane[48]. The PLpro ICP construct also contains an mEmerald with a C-terminal NLS, followed by the ZIKV NS2B, and PLpro active site coding region

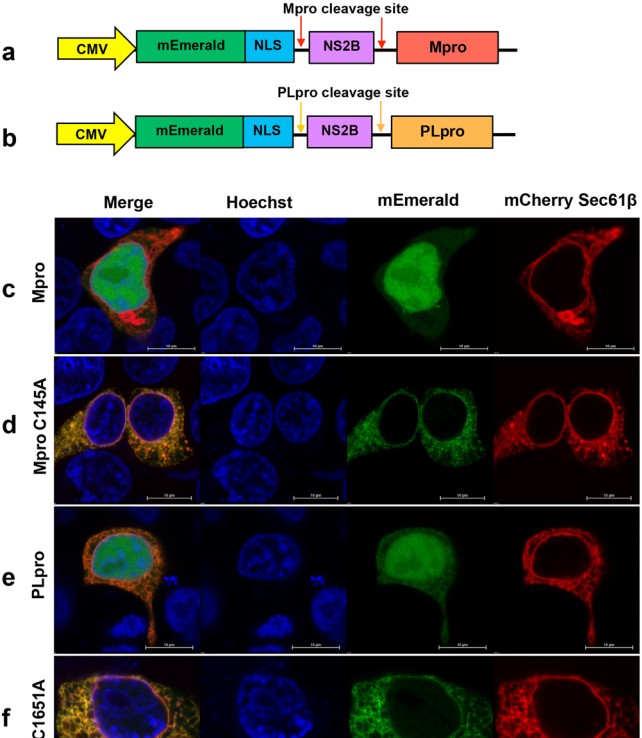

**Fig. 1 In-cell protease (ICP) assay for screening inhibitors of SARS-CoV-2 proteases. a, b** Constructs designed for the ICP assay containing mEmerald with nuclear localization signal (NLS), cleavage site for Mpro or PLpro, Zika virus NS2B followed by Mpro (**a**) or PLpro (amino acids 1541–1855 of nsp3) (**b**). **c–f** Localization of mEmerald-NLS (green) and ER marker mCherry-Sec61 β (red) at 6 h post-transfection and stained with nuclear stain Hoechst 33342 dye (blue), in ICP assay. **c** Cells transfected with ICP construct A, **d** cells transfected with ICP construct A with inactive Mpro mutant (C145A). **e** Cells transfected with ICP construct B, **f** cells transfected with ICP construct B with inactive PLpro mutant (C1651A).

(amino acids 1561–1878 of SARS-CoV-2 pp11a). The ZIKV NS2B protein is a membrane protein that anchors the protease constructs to the ER membrane. HEK293T cells at 50% confluency were co-transfected with ICP construct (Fig. 1a, b) and the ER marker mCherry-Sec61 β. When the inactive ICP constructs are expressed in cells, the mEmerald-NLS-NS2B-Mpro and mEmerald-NLS-NS2B-PLpro localize to the ER membrane due to the membrane anchoring of NS2B (Fig. 1c–f). The localization of the uncleaved mEmerald-NLS-NS2B-Mpro and mEmerald-NLS-NS2B-PLpro proteins to the ER membrane is confirmed from its co-localization with the ER marker mCherry-Sec61 β (red) (Fig. 1c–f). Upon protease cleavage, mEmerald-NLS localizes to the nucleus because of its NLS (Fig. 1c, e). The protease activity of Mpro or PLpro expressed from the construct cleaves mEmerald-NLS from the NS2B membrane anchor, and the fluorescent protein localizes to the nucleus, which was detected within 6 h post transfection (Fig. 1c, e). The inactive protease mutants, Mpro C145A and PLpro C1651A, do not cleave the mEmerald-NLS from the membrane anchor and the fluorescent protein does not accumulate in the nucleus (Fig. 1d, f). Thus, by quantifying the mEmerald fluorescence in the nucleus and total fluorescence by fluorescence microscopy, the assay specifically determines the effect of compounds inhibiting the protease activity of Mpro and PLpro in human cell lines upon treatment with protease inhibitors.

**Identification of Mpro and PLpro inhibitors using ICP assay.**
We tested a library of 64 compounds purchased from Selleckchem, including inhibitors of HIV protease, HCV protease, cysteine proteases, dipeptidyl peptidase, reverse transcriptase, and other inhibitors (Table S1) for their ability to inhibit Mpro or PLpro using ICP assay. We determined the protease inhibition of these compounds using ICP assay in HEK293T cells using different concentrations (1 and 10 μM) of compounds and collected images of live cells using confocal microscopy. The distribution of mEmerald-NLS was quantified from cells in each image ($n = 5$) using ImageJ, from the ratio of fluorescence in the nucleus and total fluorescence calculated and normalized to untreated cells. To test if the reduction in protease activity is due to cytotoxicity, we performed cell viability assays using alamarBlue on HEK293T cells after treating the cells with compounds at 10 μM concentration. From the ICP assay, we identified 11 compounds affecting Mpro and 5 compounds affecting PLpro activity, respectively (Fig. 2a). The selected compounds with inhibitory activity against SARS-CoV-2 proteases and reduced cytotoxicity compared to the untreated control were selected for antiviral studies (Fig. 2a). The selection criteria were based on a cut-off of 50% reduction in protease activity at 10 μM and 25% at 1 μM concentration of the compounds with 90% cell viability at 10 μM concentration of the compound (Fig. 2a). Confocal images from ICP assays of the cells treated with selected inhibitors show a reduced nuclear localization of mEmerald-NLS and increased localization to the ER, indicating the efficacy of compounds in inhibiting the activity of the SARS-CoV-2 proteases (Fig. 2b). From the panel of initial hits from ICP assay, we selected Daclastavir dihydrochloride and Sitagliptin as PLpro inhibitors and MG-101, Lycorine HCl, BMS-707035, Atazanavir, Lomibuvir, and Nelfinavir mesylate as Mpro inhibitors for antiviral studies against SARS-CoV-2 in BSL-3 (Fig. 3).

*Mpro and PLpro inhibitors reduce the production of SARS-CoV-2 from Huh-7.5 cells.* We next evaluated the antiviral activity of the selected PLpro and Mpro inhibitors against SARS-CoV-2 in BSL-3 using in vitro cell culture. We have determined the multiplicity of infection (MOI), and the duration of the experiments based on the growth kinetics of SARS-CoV-2 in Huh-7.5 and Vero E6 cells (Fig. S1a). The replication of SARS-CoV-2 in Huh 7.5 cells at 24 h post-infection (h.p.i.) was confirmed by immunofluorescence analyses (Fig. S1b). We tested the ability of the selected compounds to inhibit the production of SARS-CoV-2 in vitro using Huh 7.5 cells (Fig. 4). We selected Huh 7.5 cells for these experiments because they are human cells that allow a robust replication of SARS-CoV-2 to determine the levels of reduction by a compound against cells producing a high titer virus. Cell-culture supernatants were collected from the infected, drug-treated cells, and the reduction in virus titer was determined by plaque assays performed on Vero E6 monolayers. All the compounds tested were efficient in inhibiting virus replication, with $EC_{50}$ values less than 0.5 μM. The two PLpro inhibitors, Sitagliptin and Daclatasvir HCl, significantly reduced the virus titer in cell culture supernatants. Sitagliptin showed an $EC_{50}$ of 0.32 μM, $CC_{50}$ of 21.59 μM, and selectivity indices (SI) value of 67. Daclastavir HCl showed an $EC_{50}$ of 1.59 μM, $CC_{50}$ of 32.14 μM, and SI value of 20.2. Among the Mpro inhibitors, the $EC_{50}$ values ranged from 0.01 μM for Lycorine HCl to 0.038 μM for MG-101. None of the compounds exhibited significant cytotoxicity in Huh-7.5 cells, with $CC_{50}$ values ranging from 17 to 70 μM and the SI range of 67–1878. The maximum value of $CC_{50}$ was 48 μM obtained for Mpro inhibitor Lomibuvir.

*Mpro and PLpro inhibitors reduce the replication of SARS-CoV-2 in human Huh-7.5 cells.* We tested the effect of selected compounds in inhibiting the spread of SARS-CoV-2 on cell monolayers was tested by immunofluorescence assay (IFA) (Fig. 5a). From the percentage of inhibition (Fig. 5b), we calculated that the PLpro inhibitors Sitagliptin and Daclastavir inhibited virus spread by 75% and 70%, respectively, compared to untreated controls (Fig. 5b). The Mpro inhibitors MG-101 and Nelfinavir mesylate inhibited virus spread by 95%, Lycorine HCl 88%, and BMS-707035 81%, respectively (Fig. 5b). The Mpro inhibitors, Atazanavir and Lomibuvir, showed only moderate inhibition with a rate of inhibition below 50% of the untreated control. We next determined whether the reduction in virus spread in the inhibitor-treated cells correlates with virus replication. We extracted total RNA from infected cells treated with inhibitors and quantified the number of viral RNA molecules by quantitative real-time polymerase chain reaction (qRT-PCR) (Fig. 5c). The reduction in the number of RNA molecules estimated for each treatment was comparable to the reduction in the number of infected cells observed in the immunofluorescence analysis. The number of viral RNA molecules was significantly reduced in cells treated with the PLpro inhibitors Daclastavir and Sitagliptin (2-log) (2-log). Mpro inhibitors Lycorine, MG-101, and Nelfinavir mesylate each showed an approximately 3-log reduction in the number of RNA molecules compared to untreated cells. However, the reduction in the number of viral RNA molecules was lower for BMS-707035 (1-log reduction), Atazanavir (0.5-log), and Lomibuvir (0.5-log).

*Nelfinavir mesylate and Lycorine HCl do not inhibit SARS-CoV-2 entry.* Next, we tested whether the inhibitors have any secondary effects, especially against proteases involved in the processing of S protein at the host-cell membrane during entry. We selected three inhibitors, MG-101, Nelfinavir mesylate, and Lycorine HCl that showed a significant reduction in virus titer for a time of addition experiment. The pre-treatment of Huh-7.5 cells with Nelfinavir mesylate and Lycorine HCl had only minimal effect on virus production, suggesting that they do not significantly affect the entry of the virus into the cells (Fig. S2a). However, pretreatment of cells with MG-101, which is a Calpain inhibitor resulted in a 1-log reduction in virus titer. Calpain inhibitors have already been suggested to also inhibit SARS-CoV-2 entry[31].

*Combined inhibition of Mpro and PLpro has an additive effect in inhibiting the replication of the SARS-CoV-2 Delta variant.* We tested whether a combination of the most effective PLpro and Mpro inhibitors, Sitagliptin and MG-101can impart an additive antiviral effect than individual treatment. We tested a combination of 0.5 μM each of Sitagliptin and MG-101, which showed more significant antiviral activity than treatment with 1 μM of either drug for SARS-CoV-2 (Fig. 6a). Next, we tested whether the current prevalent strain of SARS-CoV-2, the delta variant, is also sensitive to the drug treatment. For this, we pretreated Huh 7.5 cells with 0.01, 0.1, and 1 μM of Sitagliptin and MG-101 or the combination of the two inhibitors and infected with SARS-CoV-2 delta variant. Compared to the inhibition with MG-101 or Sitagliptin alone, the PFU/ml of the delta variant reduced by 1 log and 2 logs at 0.1 μM and 1 μM final concentrations of MG-101 and Sitagliptin combination, respectively (Fig. 6b). We also tested whether combining two of the three inhibitors MG-101, Nelfinavir mesylate, and Lycorine HCl will have an additive effect on reducing virus titer. We treated Huh-7.5 cells with a combination of two inhibitors at 1 μM concentration each and infected with SARS-CoV-2. The combination of drugs showed additive effect and reduction in virus titer (Fig. S1b). We determined the cell viability by alamarBlue dye assay for all inhibitor treatments and the cell viability was 90-95% range compared to untreated controls (Fig. S2c).

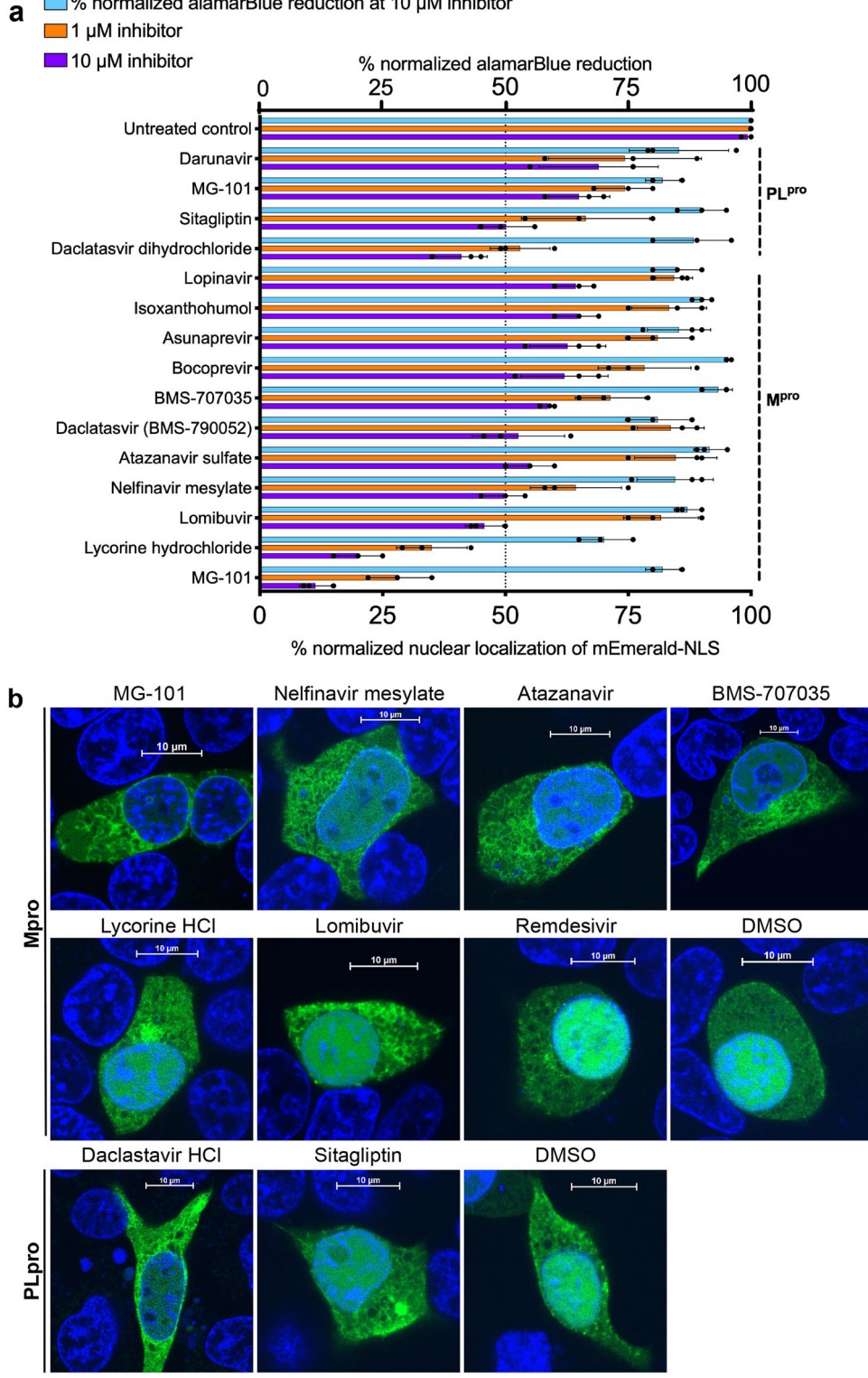

**Fig. 2 SARS-CoV-2 protease inhibitors selected after ICP assay and quantification. a** ImageJ quantification of mEmerald-NLS localization to the nucleus identifies six Mpro inhibitors and two PLpro inhibitors with approximately 50% reduction in protease activity. The selection criteria were 50% reduction at 10 μM and 25% at 1 μM concentrations without cytotoxicity. The distribution of mEmerald-NLS was quantified from cells in each image ($n = 3$) using ImageJ, the ratio between fluorescence in the nucleus, and total fluorescence calculated and normalized to that of untreated cells. **b** Live confocal images of HEK293T cells expressing PLpro or Mpro at 6 h post-transfection. Cells were treated with 10 μM inhibitors as indicated and nuclei were stained using Hoechst stain. Cells treated with DMSO and Remdesivir are negative controls.

**Fig. 3 Chemical structures of selected inhibitors with antiviral activity against SARS-CoV-2.** MG-101, Lycorine HCl, BMS 707035, Atazanavir, Lomibuvir, and Nelfinavir mesylate are inhibitors of Mpro. Sitagliptin and Daclastavir HCl are inhibitors of PLpro.

*Docking of the selected compounds to Mpro and PLpro structure.* We next performed molecular docking analysis to study potential drug-protein interactions and identify the possible ligand poses in the binding pockets of each protease. For the Mpro, a high-resolution X-ray structure in complex with the potent covalent peptidomimetic inhibitor N3 (7BQY) was used for the docking studies[49]. The protein structure analysis revealed that the catalytic dyad (Cys145/His41) is located at the cleft between two protein domains, and it is almost entirely occupied by the peptidomimetic inhibitor. By looking in detail at the protein-inhibitor interaction, the binding pocket can be divided into five regions, where the most essential features for ligand binding are present. The cleavage site is located between P1 and P1', where a covalent bond is formed through the Cβ vinyl group of N3 inhibitors and the catalytic cysteine residue Cys145 (Fig. 7a). The lactam at position P1 forms a double H-bond with His163 and Glu166, stabilizing the monomeric and inactive form of the enzyme. The P1 lactam drastically enhances the inhibitory potency due to its capacity to mimic the recognition site for glutamine, which is highly conserved in the 3 C cleavage sequence. The P2 leucine forms hydrophobic interactions in a small lipophilic cleft; meanwhile, the bulky benzyl group at the P1' interact with T24 and T25 through van der Waals interactions. The P3 and P4 have smaller contributions to the binding affinity of the N3 inhibitor. The P4 is partially solvent-exposed and makes van der Waals interact with the backbone Thr190 and Thr191 residues; instead, a solvent-exposed valine residue occupies the P3 position.

To determine the interaction of drugs with Mpro, the compounds were docked in the Mpro binding site, and the free energy of binding was calculated using Molecular Mechanics

Generalized Born Surface Area (MMGBSA) approach. Among the six tested drugs (MG-101, Lycorine, BMS-707035, Atazanavir, Lomibuvir, and Nelfinavir), MG-101 showed the highest affinity ($\Delta G = -73.68$ kcal/mol). Although the MG-101 is a smaller peptidomimetic than N3, it showed some crucial interaction, such as the Val in P3 and a reactive aldehyde group, forming a covalent bond with Cys145. To further analyze the Cys145 thiol nucleophilic attack on the aldehyde group, the CovDock approach, implemented in Maestro, was used to optimize the binding poses and predict the covalent bond formation. The results showed that the MG-101 could easily bind the cysteine group, maintaining the hydrophobic interaction of the Val in P3. Moreover, the presence of several hydrogen bonds with the surrounding residues might be essential for the correct positioning of the compound into the pocket (Fig. 7b1). Atazanavir showed the second-best affinity ($\Delta G = -68.13$), but in contrast to MG-101, not all of the docked poses showed a good score, with a $\Delta G$ value between $-34.22$ and $-52.31$ kcal/mol. These differences can be due to the presence of a rigid biphenyl group, which cannot be easily accommodated in the binding pocket without exposing other hydrophobic groups to the solvent (Fig. 7b2). Moreover, although in most of the binding poses, Atazanavir did not appear to interact with His41 or Cys145, this compound could fully occupy the active site, which likely explains the good MMGBSA score. Lomibuvir also presented highly variable results in terms of occupation of the binding site, mostly due to the presence of the triple bond, which reduces compound flexibility, increasing the ligand exposure to the solvent. However, interestingly, few poses showed a hydrogen bond interaction between the acid group and the His163, and the cyclohexene group is inserted

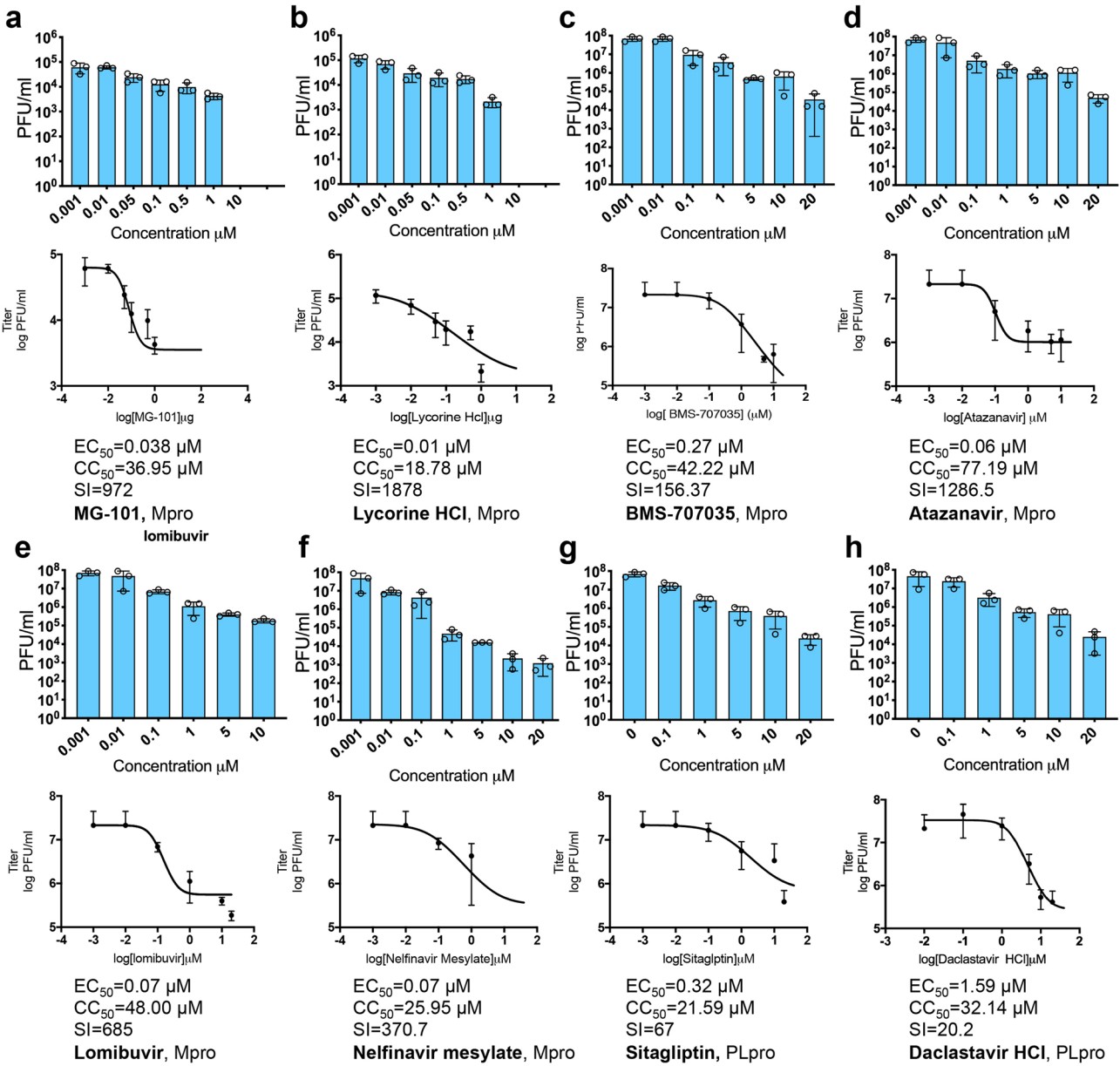

**Fig. 4 Inhibition of SARS-CoV-2 replication in Huh-7.5 cells by compounds selected from ICP assay.** Reduction in virus titers for different concentrations of compounds against Mpro (**a–f**) and PLpro (**g**, **h**) was determined by plaque assays ($n = 3$). Dose-response curves were plotted, $EC_{50}$ (50% Effective concentration) was determined from plaque assay and $CC_{50}$ (50% cytotoxic concentration) was determined by alamarBlue reduction data. Selectivity index $SI = CC_{50}/EC_{50}$.

deeply in the hydrophobic site (Fig. 7b3). Unlike Atazanavir and Lomibuvir, the binding model of Nelfinavir showed that the compound could be accommodated well within the active sites. However, in this case, only a few interactions were noted, including three hydrogen bonds with Glu166 and Asn142 and the possible formation of a π-π stacking interaction with His41 (Fig. 7b4). Lycorine HCl and BMS-707035 are smaller compounds than Atazanavir and Nelfinavir, which can only partially occupy the active site (Fig. 7c1, c2, respectively). BMS-707035 can form a hydrogen bond between the carbonyl group and His163, and the fluorobenzyl group could fit well within the hydrophobic site (P2), whereas the Lycorine HCl binding appeared to rely primarily upon hydrogen bond interactions with the surrounding polar residues Asn142, Gln189, and Glu166.

We used the crystal structures of SARS-CoV-2 PLpro in complex with covalently bound peptide inhibitor (PDB ID

6WX4) to evaluate the binding affinity of Daclastavir and Sitagliptin with PLpro[50]. The co-crystallized peptide inhibitor, VIR251, forms a covalent bond with the catalytic Cys111, through a Michael addition reaction. The covalent bond is facilitated by a large number of hydrogen bond interactions with the polar residues Gly163, Tyr268, Gly271, Trp106, Asp164, and Tyr264 (Fig. 7d). The docking studies showed that Sitagliptin binds adjacent to the active site, blocking the entrance to the catalytic triad (Cys111, His272, and Asp286). The binding interaction is stabilized by the trifluorobenzyl ring, which forms hydrophobic interaction with the side chains of the prolines, Tyr264, Pro248, Pro247, and Thr301 (Fig. 7e1). Interestingly, the binding mode of Sitagliptin appears to be similar to that of a known SARS-CoV inhibitor 3k (PDB; 4OW0)[35]. SARS-CoV-2 PLpro has a high similarity with the corresponding SARS-CoV protease (82.80% sequence similarity), and amino acids in the active site are highly

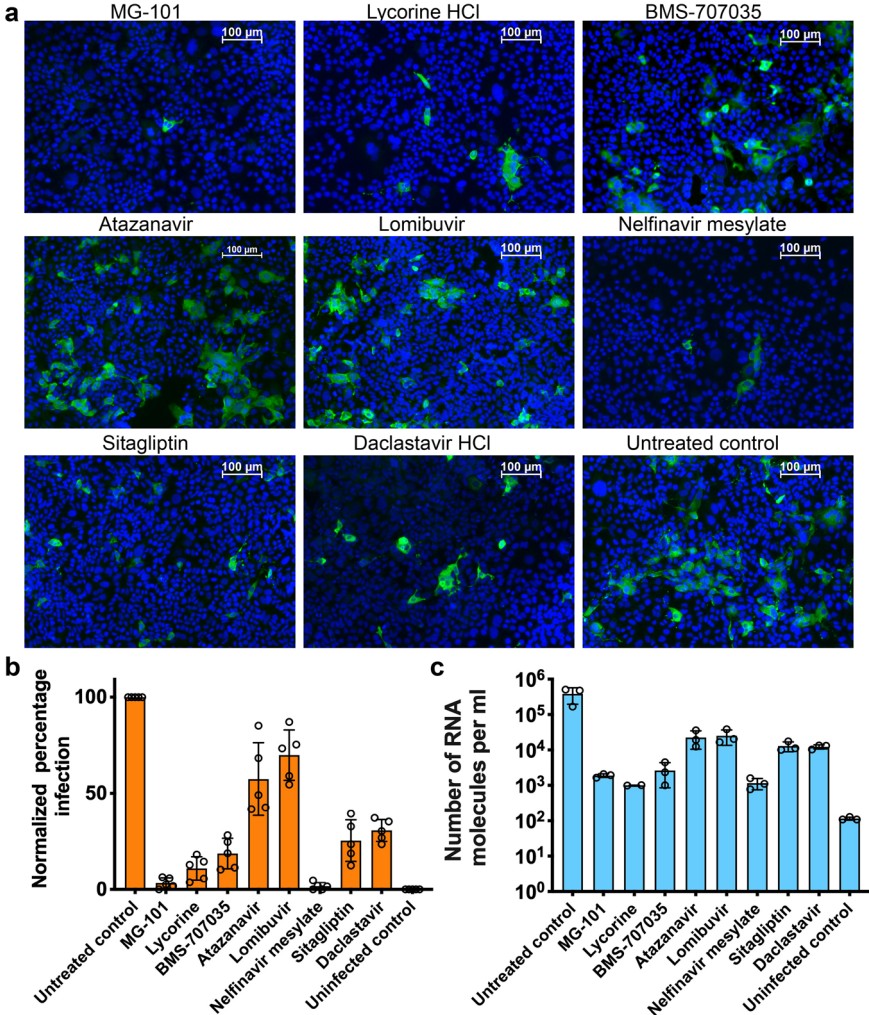

**Fig. 5 Effect of compounds selected from ICP assay on SARS-CoV-2 replication. a** Huh-7.5 cells positive for SARS-CoV-2 determined by immunofluorescence assay. Huh-7.5 cells pre-treated for 24 h with inhibitors were infected with SARS-CoV-2 at an MOI of 0.1. At 24 h.p.i., cells were fixed with paraformaldehyde and probed with anti-N (SARS CoV-2 nucleocapsid) primary antibody and FITC-conjugated goat anti-rabbit secondary antibody (green, infected cells). Nuclei were stained with Hoechst stain and confocal images were acquired. **b** The ratio of infected to uninfected cells for each treatment was calculated using Nikon Elements software and normalized to untreated controls ($n = 5$). **c** Quantification of SARS-CoV-2 RNA molecules from infected cells treated with inhibitors as indicated ($n = 3$).

conserved (Pro248, Pro249, Tyr269, Asp165, Glu168, Leu163, Gly164, Gln270, Tyr274, Try265, and Thr302). For this reason, the binding mode of Sitagliptin was compared with the co-crystallized SARS-CoV inhibitor. The protein superposition of the two crystal structures revealed that the trifluorobenzyl ring of Sitagliptin occupied the same position as the 1-naphthyl rings in the hydrophobic site of the pocket. Moreover, an additional, important similarity was also observed between the piperidine ring nitrogen and the amino group of the Sitagliptin, with both of these interacting with the side-chain carboxylate of Asp165 (Fig. 7e2). The geometry of the PLpro is characterized by a very narrow catalytic site, which requires a highly flexible compound, limiting the binding of a more rigid and large molecule, such as Daclatasvir. The docking results show that the Daclatasvir cannot insert in the narrow active site of the protease, and most of the compound is exposed to the solvent, which negatively influences the ΔG score of the compounds (Fig. 7f).

**In vitro inhibition of Mpro activity by MG-101.** FRET assay using Dabcyl–Edans fluorescence pair was performed to validate the inhibitory potential of the ICP assay screened Mpro

inhibitors. Among them, compound MG-101 (Fig. 8a) had a micromolar ($2.89 \pm 0.86\,\mu M$) half-maximal inhibitory concentration ($IC_{50}$) (Fig. 8c). Compounds Lycorine, BMS-707035, Atazanavir, Lomibuvir, and Nelfinavir were not found to be effectively inhibiting the Mpro protease activity. GC376 (Fig. 8b) was used as a reference and showed an $IC_{50}$ value of $0.13 \pm 0.07\,\mu M$, consistent with a reported value of $0.19\,\mu M$[29].

**In vitro inhibition of PLpro activity by Daclastavir-HCl and Sitagliptin.** FRET assay was performed to determine the efficacy of screened inhibitors against PLpro protease. The compounds Daclastavir-HCl (Fig. 8d) and Sitagliptin (Fig. 8e) showed $IC_{50}$ values of $1.838 \pm 0.256\,\mu M$ and $1.138 \pm 0.19\,\mu M$, respectively.

**X-ray crystal structure determination of the Mpro in complex with MG-101.** To reveal the detailed binding interaction and the mechanism of Mpro inhibition by MG-101, we crystalized the Mpro and MG-101 complex and determined its high-resolution X-ray crystal structure. Firstly, crystallization conditions for the apo-form Mpro were screened using commercially available crystallization solutions. Clusters of thin plate-like crystals were

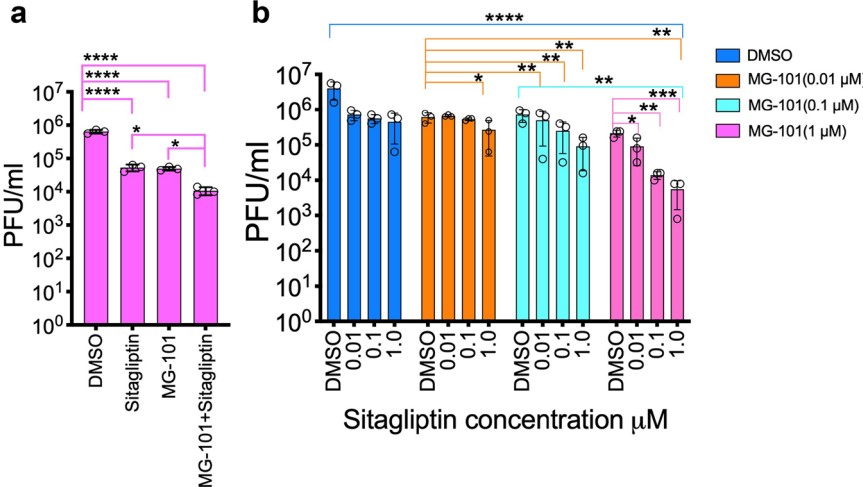

**Fig. 6 Antiviral activity of combinations of Mpro and PLpro inhibitors.** Effect of combination of Mpro inhibitor MG-101 and PLpro inhibitor Sitagliptin on the growth of SARS-CoV-2 (**a**) and the delta variant (**b**) ($n = 3$). Huh7.5 cells were treated with the inhibitors as shown, and reduction in virus titer at 24 h.p.i. was determined by plaque assays. Data are shown as mean ± SEM. $p$ Values were considered significant when $p < 0.05$ (*), $p < 0.01$ (**), $p < 0.001$(***), and $p < 0.0001$(****).

**Table 1 X-ray crystallographic data collection and refinement statistics.**

|  | Apo-form 1 | Apo-form 2 | Mpro - Calpain I |
|---|---|---|---|
| PDB code | 7LKE | 7LKD | 7LBN |
| *Data collection* |  |  |  |
| Wavelength | 0.9687 | 0.9687 | 1.130 |
| Space group | C2 | P21 | C2 |
| a, b, c (Å) | 111.303, 53.227, 44.381 | 44.307, 53.703, 113.933 | 97.481, 80.729, 51.579 |
| α, β, γ (°) | 90, 101.756, 90 | 90, 101.012, 90 | 90, 90, 90 |
| Resolution (Å) | 50.00–2.70 (2.80–2.70)[a] | 50.00–2.00 (2.03–2.00)[a] | 50.00–1.76 (1.79–1.76)[a] |
| $R_{merge}$ (%) | 10.4 (69.5)[a] | 8.9 (94.6)[a] | 3.8 (29.5)[a] |
| I / σI | 11.6 (1.4)[a] | 21.5 (1.6)[a] | 38.4 (3.9)[a] |
| Completeness (%) | 92.8 (76.3)[a] | 99.8 (99.9) | 97.0 (72.4)[a] |
| Redundancy | 2.9 (2.5)[a] | 4.3 (3.9)[a] | 3.2 (2.4)[a] |
| CC1/2 | 0.982 (0.531)[a] | 0.993 (0.474)[a] | 0.995 (0.905)[a] |
| *Refinement* |  |  |  |
| Resolution (Å) | 50.00–2.70 (2.79–2.70)[b] | 50.00–2.00 (2.08–2.00)[b] | 50.00–1.76 (1.83–1.76)[b] |
| No. of reflections | 6591 (472)[b] | 35,074 (3316)[b] | 35,055 (3162)[b] |
| $R_{work}/R_{free}$ (%) | 22.5/29.3 | 18.2/22.6 | 16.6/19.8 |
| *No. of atoms* |  |  |  |
| Protein | 2323 | 4716 | 2332 |
| Ligand | – | – | 63 |
| Water | 0 | 234 | 310 |
| Protein residues | 300 | 610 | 301 |
| B factors (Å²) | 76.65 | 37.02 | 32.13 |
| *r.m.s. deviations* |  |  |  |
| Bond lengths (Å) | 0.010 | 0.007 | 0.008 |
| Bond angles (°) | 1.48 | 1.01 | 0.99 |
| *Ramachandran plot* |  |  |  |
| Favored (%) | 96.98 | 97.19 | 97.32 |
| Allowed (%) | 2.35 | 2.48 | 2.68 |
| Outliers (%) | 0.67 | 0.33 | 0.0 |
| Clashscore | 10.01 | 6.97 | 5.17 |
| No TLS groups | 3 | 6 | 3 |

[a]Highest resolution shell is shown in parenthesis.
[b]Highest resolution shell is shown in parenthesis.

observed after 4 days in various conditions of the PACT premier screen (Molecular Dimensions). High-resolution diffraction quality crystals were obtained by using crystallization solution containing 0.2 M sodium sulfate and 10% PEG 3350, and the X-ray structures of apo-form Mpro were determined in two different space groups, C2 and P21, comprising of one and two Mpro protomers in their asymmetric units, respectively (Table 1). These structures are similar to other Mpro crystal structures available in PDB. Mpro and MG-101 complex preparation was attempted by soaking Mpro crystals into MG-101 solution, but it was not successful. We, therefore, formed the Mpro and MG-101 complex by prolonged pre-incubation followed by crystallization screening. Crystallization conditions were optimized to form good diffracting rectangular-shaped crystals. The crystal structure of Mpro and MG-101 complex was determined at 1.76 Å resolution as a C2 space group containing two Mpro protomers in an asymmetric unit (Fig. 9a, Table 1).

**Structure of the SARS-CoV-2 Mpro and MG-101 complex.** The structure of the Mpro and MG-101 complex shows unambiguous electron density for the inhibitor located at the active site (Fig. 9b). Continuous electron density between the sulfhydryl group of catalytic Cys145 residue of Mpro and the aldehyde warhead of MG-101 shows a formation of thiohemiacetal covalent bond, indicating that it inhibits the Mpro activity by preventing the substrate binding at the active site. The inhibitor is accommodated within the S1–S3 sites of the substrate-binding channel using its 503.9 Å² of the solvent-accessible surface and established hydrophobic and hydrophilic interactions with the catalytic and substrate binding residues (Fig. 9c, Fig. S3). The P1 leucine side chain of MG-101 protrudes into the S1 subsite where main chain oxygen of His164 stabilizes an amide backbone of inhibitor by hydrogen bonding. The P2 leucine side chain of inhibitor fits in the hydrophobic S2 pocket of Mpro through interactions with His41, His164, Met165, and Asp187 side chains, and the P3 leucine side chain of inhibitor occupies the solvent-accessible substrate-binding site S3 of Mpro through H-bonding with Glu166 side chain. The acetyl moiety at the P3 position further affirms the positioning of the inhibitor in the substrate-binding site through hydrophobic interactions with Glu166 and Gln189 side chains of Mpro. The hydroxyl moiety of thiohemiacetal structure from the inhibitor occupies the oxyanion hole

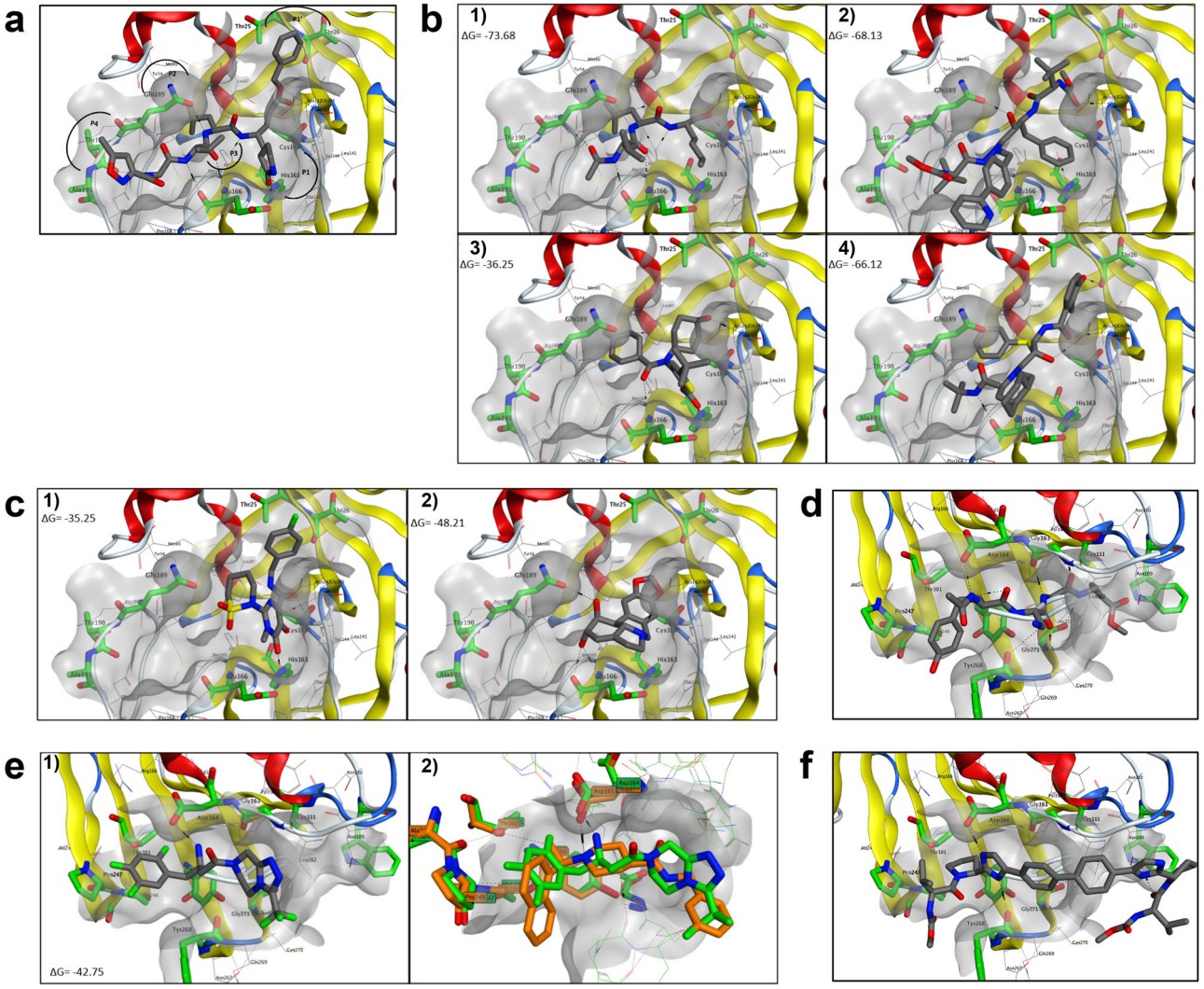

**Fig. 7 Docking of SARS-CoV-2 inhibitors.** The key residues forming the binding pocket are highlighted in green. H-bonds are depicted as dashed black lines. **a** Co-crystallographic pose of peptidomimetic inhibitor N3 (PDB 7BQY) in the active site of SARS-CoV-2 Mpro. **b** Predicted binding mode for (1) MG-101, (2) Atazanavir, (3) Lomibuvir, and (4) Nelfinavir in the Mpro- active site. **c** Predicted binding mode for (1) BMS-707035, and (2) Lycorine HCl in the Mpro- active site. **d** Co-crystallographic poses of VIR251 (PDB 6WX4) in the active site of SARS-CoV-2 PLpro. **e** (1) Predicted binding mode for Sitagliptin in the PLpro-active site, (2) Superposition of SARS-CoV crystal structure (4OW0, orange) and SARS-CoV-2 crystal structure (PDB 6WX4, green) the docked Sitagliptin. **f** Predicted binding mode for Daclatasvir in the PLpro- active site.

of Mpro formed by the backbone amide groups of Gly143 and Cys145. Similar to other aldehyde-containing peptidomimetic inhibitors like GC376 and calpain inhibitor II, the thiohemiacetal of MG-101 adopts the (S)-isomer conformation, which is the result of a nucleophilic attack by Cys145 of Mpro onto the planar carbonyl of inhibitor[29,51]. Consistent with the (S) configuration, an imidazole of His41 does not form any hydrogen bond with the thiohemiacetal oxygen; instead, this carbonyl oxygen of inhibitor points away and goes into an oxyanion hole where it is stabilized by hydrogen bonds formed with the main chain amides of Gly143 and Cys145.

**Conserved mode of binding of peptidomimetic covalent inhibitors at the Mpro active site**. To understand how peptidomimetic inhibitor binds at the catalytic site of the SARS-CoV-2 Mpro and obtain the structure-activity relationship, we compared the crystal structures of the Mpro in complex with MG-101 (this study) and with GC376 (PDB 6WTJ) (Fig. 9d). Both inhibitors interact in an almost similar fashion in the S1–S3 substrate binding sites of Mpro. With a few variations—His163 (GC376)

and Gly143 (MG-101)—the same set of the Mpro active site residues is involved in hydrogen bonding with these inhibitors. P2 leucine of GC376 fits in the hydrophobic S2 subsite akin to the MG-101, whereas the P3 is exposed to solvent. Due to the P3 acetyl group interaction in the S3 subsite, MG-101 is positioned at the Mpro active site in extended conformation compared to GC376. Although $IC_{50}$ of MG-101 is 21 times higher than $IC_{50}$ of GC376, the buried surface of MG-101 (503.8 Å$^2$) is larger than the one of GC376 (404.2 Å$^2$). We speculate that the glutamine surrogate at the P1 position of GC376 lowers its $IC_{50}$ value. Besides these aberrations, the overall mode of fitting of both the inhibitors shows a conserved mode of binding and interactions in the substrate-binding site of the SARS-CoV-2 Mpro.

**Discussion**
The continuing threat to global health posed by the SARS-CoV-2 and its variants with increased abilities to spread and escape from immunity demands an arsenal of approaches and drug modalities that should likely include small molecules as antiviral agents. Repurposing approved pharmaceutical drugs provides an

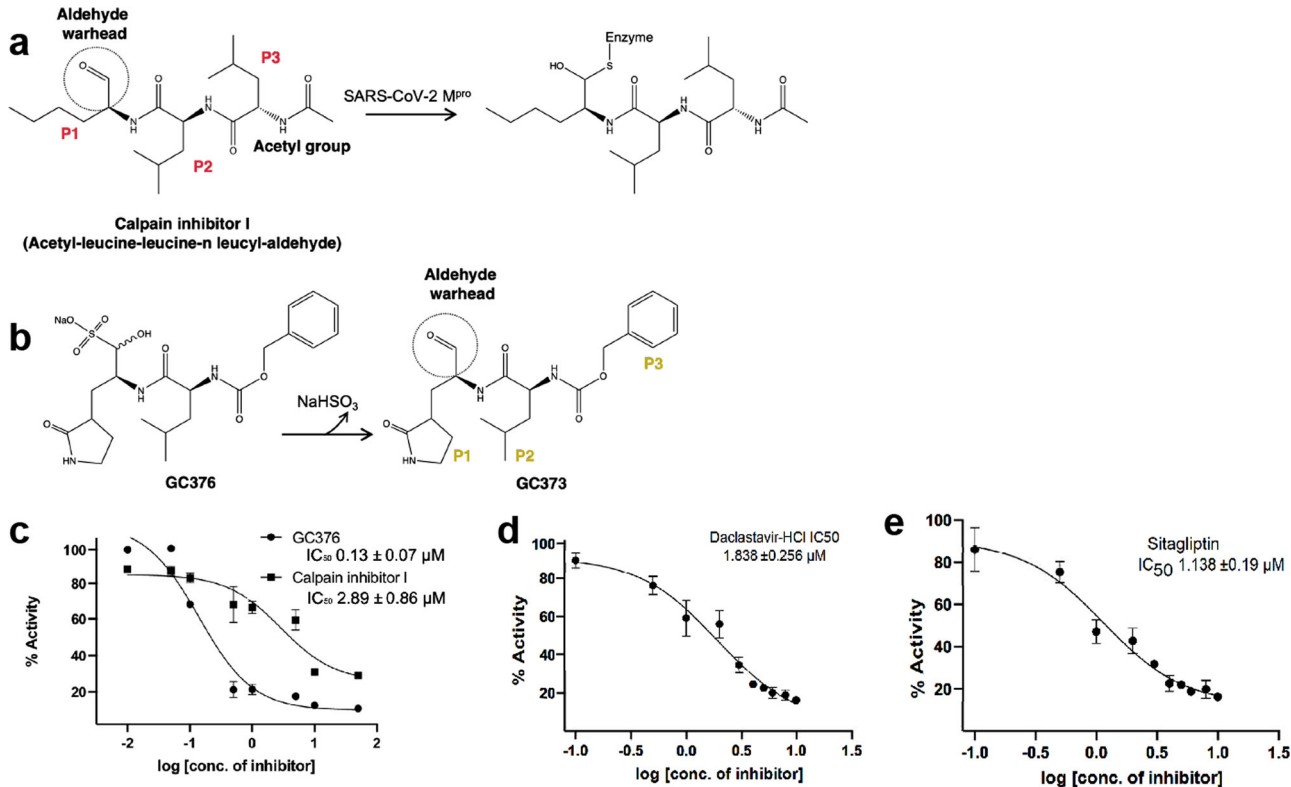

**Fig. 8 Inhibitions of the Mpro and PLpro activities in vitro by screened compounds. a** Schematic representation of the inhibition of Mpro by MG-101. Chemical groups and substrate positions (P1–P3) are indicated. **b** Chemical structures of GC376 (prodrug) and GC373 (active drug). **c** $IC_{50}$ values of MG-101 and GC376 for the cleavage of (Dabcyl)-KTSAVLQ*SGFRKME(Edans) substrate by Mpro. A cleavage site is indicated by an asterisk. The inhibitory activities of **d** Daclastavir-HCl and **e** Sitagliptin against PLpro were tested using fluorogenic peptide Z-RLRGG-AMC as the substrate. $N = 3$, values are represented as mean ± SE.

alternative approach that allows for the rapid identification of potential drugs to combat COVID-19. Although the drug repurposing approach only screens approved drugs optimized for absorption, distribution, metabolism, excretion, and pharmacokinetics (ADME-PK), an approach explicitly targeting SARS-CoV-2 proteases expressed in cells has several advantages. First, the rapid development of effective antivirals for clinical use against emerging viruses such as SARS-CoV-2 is exceedingly challenging due to the delay in conventional drug development that takes years of research and cost billions of dollars[13]. Second, among the different therapeutic targets for antiviral drug development, proteases demonstrated to be generally druggable, and as a result, they should be a priority for inhibitors of SARS-CoV-2. Third, though several biochemical assays exist that could enable high-throughput screening (HTS), cell-based assays of SARS-CoV-2 protease inhibition have not been reported previously. Cell-based assays may also provide important advantages as they factor in cell permeability, stability, and cytotoxicity.

To evaluate the potential of our cell-based assay as a tool to interrogate compound libraries and identify protease inhibitors, we conducted a focused screen using a selection of approved drugs and identified several bona fide inhibitors that are effective against SARS-CoV-2. Furthermore, most large-scale drug screening assays have been performed in silico or in vitro using purified proteins and in vivo assay that scores for virus-induced cytopathic effects[9,52]. Thus, the cell-based phenotypic screening is a feasible approach that is compatible with high-throughput pipelines, and it can identify the molecular target or mechanism of action in BSL-2[13,49]. Compared to cell-free biochemical assays, the ICP assay we developed here can be used to manually screen 100 s of compounds that target proteases in a more physiological

setting. Moreover, this assay has the potential of being developed into HTS format to screen 1000 s of compounds using an automated image-based high-throughput screening[53]. From a set of 64 repurposed drugs selected for their protease inhibition properties, we identified 16 inhibitors using the ICP assay, including 11 compounds inhibiting Mpro activity and five compounds inhibiting PLpro activity. After prioritizing compounds based on inhibitory activity against the target protease combined with low cellular toxicity, six inhibitors for Mpro (MG-101, Lycorine HCl, BMS-707035, Atazanavir, Lomibuvir, and Nelfinavir mesylate) and two inhibitors for PLpro (Sitagliptin and Daclastavir) were selected.

We found that Sitagliptin effectively inhibits PLpro activity and decreases the replication of SARS-CoV-2 in Huh-7.5 cells with an $EC_{50}$ of 0.32 μM and a $CC_{50}$ of 22 μM and SI of 67 (Fig. 4), whereas it does not inhibit Mpro activity (Fig. 2). Sitagliptin is an FDA-approved, highly selective dipeptidyl peptidase-4 inhibitor used to treat Type 2 diabetes. A recent study has found that in type 2 diabetes patients hospitalized with COVID-19, those who had Sitagliptin added to the standard care of diabetes treatment (insulin administration), the mortality rate was lowered (18% vs. 37% of deceased patients)[54]. This group had improved clinical outcomes (60% vs. 38% of improved patients; $P = 0.0001$) and with a greater number of hospital discharges (120 vs. 89 of discharged patients; $P = 0.0008$) respectively, compared with patients receiving standard of care. Interestingly, it was hypothesized that the Sitagliptin could be effective against SARS-CoV-2, blocking proteins essential for the viral entry, such as DPP-4/CD26, which was already identified as binding partners for corona-like viruses to enter host cells[20,55]. However, our docking studies showed that Sitagliptin could effectively bind the PLpro,

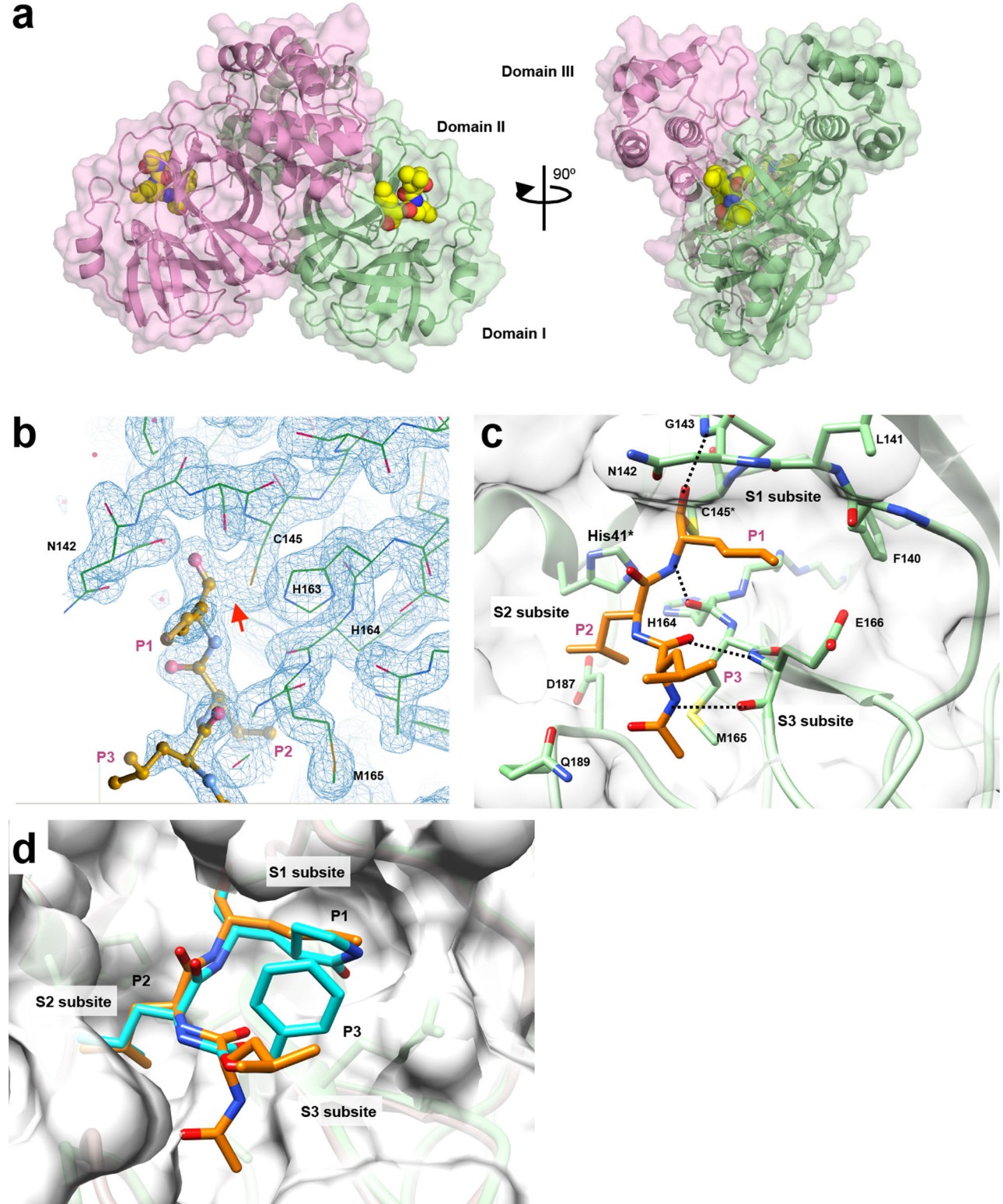

**Fig. 9 Structural basis for the Mpro inhibition. a** Crystal structure of the Mpro and MG-101 complex. Mpro dimer is depicted as a cartoon model with a transparent surface and each protomer is colored green and pink. MG-101 is shown as CPK representation. **b** Electron density map (*2Fo–Fc*, blue mesh) of MG-101 bound at the active site of Mpro. The inhibitor and Mpro are depicted as ball-and-stick and wire models, respectively. A thiohemiacetal covalent between the inhibitor and C145 residue of Mpro is indicated by a red arrow. **c** Interactions between the inhibitor and active site of Mpro. The binding of MG-101 is stabilized by H-bonds (black dot lines) with Gly143, Cys145, His164, and Glu166. **d** Comparison of the binding of MG-101 (orange) and GC373 (cyan) at the active site of Mpro. S1–S3 subsites of Mpro and P1–P3 of inhibitors are indicated.

blocking the entrance to the catalytic site, similar to other well-known SARS-COV PLpro inhibitors. These observations are in line with the biological results and indicate that Sitagliptin might facilitate the recovery of COVID-19 patients by inhibiting SARS-CoV-2 PLpro, which has not been previously reported. The second PLpro inhibitor, Daclatasvir HCl, is an FDA-approved inhibitor of HCV NS5A[56]. Our study suggests that Daclastavir HCl can decrease the activity of PLpro, leading to the reduction in SARS-CoV-2 replication in Huh-7.5 cells. Since we did not detect a reduction in Mpro activity in cells treated with Daclastavir HCl, PLpro inhibition seems specific. However, our docking studies indicate that Daclatasvir has low binding affinities to PLpro active site due to its large size and sterically hindering groups. These results suggest that the Daclatasvir might show its inhibitory effect by binding at an allosteric site instead of the active site of the enzyme.

In this study, we found that Lycorine HCl, Atazanavir, Nelfinavir mesylate, Lomibuvir, and MG-101 inhibit replication of SARS-CoV-2 in Huh-7.5 cells by specifically inhibiting Mpro activity (Fig. 4b). Among the Mpro inhibitors, the most potent drug was Lycorine HCl with an $EC_{50}$ of 0.01 µM. The drug has a $CC_{50}$ 19 µM and SI of 1878. Lycorine is an active alkaloid abundant in plants belonging to *Amaryllidaceae* with a wide range of biological functions for cancer and infectious diseases treatment[57]. Lycorine has been reported to inhibit SARS-CoV in Vero E6 cells with an $EC_{50}$ of 16 nM and SI of approximately 900 and SARS-CoV-2 in Vero E6 cells with $EC_{50}$ of 300 nM and SI of 130 although, the mechanism of inhibition has not been determined. Lycorine has been shown to be effective against human enterovirus EV71, by inhibiting the elongation of the viral polyprotein during translation and by downregulating autophagy[58,59]. Lycorine has antiviral activity against the avian influenza virus by causing the retention of the ribonucleoprotein complexes within the nucleus and against flaviviruses such as West Nile Virus and dengue virus in Vero E6 cells but the exact mechanism is unclear[60,61]. Our docking results indicate that Lycorine HCl could bind in the active site of Mpro and can form a series of hydrogen bond contacts with the surrounding polar residues Asn142, Gln189, and Glu166. Although other in silico studies also suggested a potential interaction of Lycorine HCl with the main protease with a good binding affinity, the lack of interaction with key residues in the pocket might suggest that the drug could show its antiviral activity interacting with other important targets involved in the viral replications, such as SARS-CoV-2 RdRP. Atazanavir (ATV) is an HIV protease inhibitor, which can inhibit SARS-CoV-2 replication, alone or in combination with ritonavir (RTV) in Vero E6 cells with an $EC_{50}$ of $0.5 \pm 0.08$ µM and $2.0 \pm 0.12$ µM, respectively[62]. Atazanavir was previously identified as a SARS-CoV-2 Mpro inhibitor by docking and molecular dynamic studies. It was reported that the biphenyl group Atazanavir could interact with the residues Met49, Pro52, and Tyr54 through hydrophobic interactions, and the nitrogen atoms of carbamate groups can form hydrogen bonds with residues Cys145 and Glu166, exposing the hydrophobic isopropyl group to the solvent[63]. Fintelman-Rodrigues and colleagues have suggested that Atazanavir binds Mpro by the steric occupation of the cleft in the enzymatic active site aided by the formation of hydrogens bonds with the amino acid residues Asn142 and His164[62]. These results agree with our docking studies, confirming the good binding affinity of Atazanavir versus the Mpro. However, the highly variable binding poses also indicate that the drug could not perfectly match the binding site of the protein. Nelfinavir mesylate, an inhibitor of HIV protease, has also been reported to be a potent inhibitor of spike protein-mediated virus fusion as the drug could prevent syncytia formation in Vero E6 cells expressing SARS-CoV-2 Spike protein[64]. Nelfinavir showed the highest

binding affinity among 20 drugs screened against Mpro using molecular docking[65]. Similar results have been reported by a recent study where 30 drugs have been evaluated as potential Mpro inhibitors using two binding free energy calculations, mm/gbsa, and SIE[65,66]. In both methods, Nelfinavir was identified as the most promising compound with predicted binding free energies of $-24.69 \pm 0.52$ kcal/mol by MM/GBSA and $-9.42 \pm 0.04$ kcal/mol by SIE, respectively. Together with our ICP assays and in silico studies suggest that Nelfinavir can form complexes with Mpro with good binding affinities and place it as one of the top candidates inhibiting SARS-CoV-2 with an $EC_{50}$ of 0.07 µM and a $CC_{50}$ of 26 µM.

Among the Mpro inhibitors affecting virus replication, the reduction in the number of viral RNA molecules in infected cells was less prominent for BMS-707035, Atazanavir, and Lomibuvir (0.5–1-log reduction) compared to Lycorine, Nelfinavir, and MG-101 (3-log reduction). Lomibuvir and BMS-707035 are two small molecules, which have not been previously identified as Mpro inhibitors. Lomibuvir is a non-nucleoside reverse transcriptase inhibitor that is also known to inhibit HCV RdRP. We found that Lomibuvir reduced SARS-CoV-2 production with an $EC_{50}$ of 0.07 µM and a $CC_{50}$ of 48 µM. BMS-707035 is a known inhibitor of HIV integrase and decreased SARS-CoV-2 production with an $EC_{50}$ of 0.27 mM and a $CC_{50}$ of 42.22 µM. Our in silico studies suggest that although the two molecules exhibit a lower binding affinity compared to other screened drugs, they were able to make H-bond interaction with the key residue His163 in the pocket. Moreover, several surrounding residues could form H-bonds with both drugs (His41, Gly143, Asn142, Cys145, and Glu166), leading to the stabilization of the protein–drug complex. These observations indicate that the drugs have the potential to bind within the active site of Mpro albeit with lower affinity, in agreement with having lower efficacy as SARS-CoV-2 inhibitors.

MG-101 is a potent inhibitor of cysteine proteases, which inhibits calpain I, calpain II, cathepsin B, and cathepsin L. A previous study screening calpain inhibitors against Mpro using a thermal shift assay has already identified MG-101 as an inhibitor of SARS-CoV-2 with an IC50 of 8.6 µM[51]. Similar to this finding, using our ICP assay, we have also confirmed that MG-101 as a potent inhibitor of Mpro and further virus reduction assays showed MG-101 inhibits SARS-CoV-2 replication in Huh7.5 cells with an $EC_{50}$ of 0.038 µM. It has been reported that calpain inhibitors II and XII inhibit SARS-CoV-2 in the CPE assay with $EC_{50}$ values of 2.07 and 0.49 µM, respectively by inhibiting Mpro activity[51]. The structures of Mpro in complex with calpain inhibitors II and XII that are structurally dissimilar to the traditional Mpro inhibitors GC-376 have been solved by X-ray crystallography[31]. While the structure of Mpro bound to calpain inhibitor II revealed a canonical, extended conformation, in contrast, the calpain inhibitor XII adopts an atypical binding mode. The calpain inhibitor XII was found to bind Mpro active site with an inverted, semi-helical conformation placing the P1′ pyridine ring instead of the P1 norvaline side chain in the S1 pocket[31]. The crystal structure of the Mpro and MG-101 complex show that MG-101, which is calpain inhibitor I, binds to the active site and the binding is stabilized by H-bonds with Gly143, Cys145, His164, and Glu166, which presumably contributes to its strong inhibitory activity against SARS-CoV-2 replication with an $EC_{50}$ of 0.038 µM. Comparison of the structures of Mpro bound to MG-101 with similar aldehyde inhibitors like MG-132 shows a conserved mode of binding of the peptide-based inhibitors in the protease active site[67]. To test whether MG-101 also affects viral entry by inhibiting protease processing of the spike protein, we performed a time of addition experiment. We found that when the cells were pretreated with the selected inhibitors, only MG-101 has shown to affect the virus entry, suggesting that the pre-

treatment of cells with MG-101 also affects the initial processes involved in the virus entry. Calpain and cathepsin inhibitors such as MDL28170 (calpain inhibitor III) have been shown to inhibit SARS-CoV replication in Vero E6 cells as well as inhibit cellular proteases necessary to fully activate the viral glycoprotein's membrane-fusion potential[68]. Our results from the mechanism of MG-101 inhibition of SARS-CoV-2 thus support the prospect of calpain inhibitor structure-based drug designing of new dual-inhibitors targeting both the Mpro and host cathepsin L for generating effective SARS-CoV-2 antivirals[31,69].

Drug combination therapy is an effective treatment against viruses like HIV and HCV[70,71]. Combining drugs with discrete targets against the same disease or agent can achieve more potent therapeutic effects, decrease the required dose, thereby reducing side effects. We reasoned that combining the two inhibitors we identified, one each for Mpro and PLpro would have better anti-SARS-CoV-2 results than treating with individual Mpro or PLpro inhibitors. Combinations of Mpro inhibitor MG-101 and PLpro inhibitor Sitagliptin improved the antiviral effect on the growth of SARS-CoV-2 Delta variant at concentrations 0.1 to $-1\,\mu M$ (Fig. 6b). Furthermore, MG-101 showed significantly improved antiviral activity against SARS-CoV-2 when cells were treated with other Mpro inhibitors Lycorine HCl or Nelfinavir at $1\,\mu M$ concentration each (Fig. S2b). A combination of MG-101 and Lycorine HCl or Nelfinavir mesylate showed a 3–4 log reduction in virus titer at $1\,\mu M$ concentration of each drug compared to untreated cells. Together, our results suggest that the combined inhibition of Mpro and PLpro or a combination of different drugs targeting the same protease is an attractive avenue for therapy against COVID-19.

In summary, the ICP assay described here can be used to rapidly screen effective protease inhibitors. The eight protease inhibitors identified in this study as bona fide SARS-CoV-2 inhibitors from a library of 64 repurposed drugs, two targeting PLpro and 5 targeting Mpro, can be optimized to develop therapeutics for COVID-19. The PLpro inhibitor Daclastavir HCl, Mpro inhibitors Lomibuvir, and BMS-707035 reported here are new inhibitors of SARS-CoV-2. The structure of the most effective inhibitor MG-101 reported here is a calpain inhibitor with a very similar structure to the already reported calpain inhibitors of SARS-CoV-2[29,51]. Through ICP assays and docking studies, previously reported SARS-CoV-2 inhibitors Sitagliptin has been identified here as a PLpro inhibitor and Lycorine HCl and Nelfinavir mesylate as Mpro inhibitors. Finally, the combined inhibition of Mpro and PLpro drugs reported here might serve as a suitable therapeutic inhibition strategy to combat COVID-19. Future work will be required to develop new effective therapeutics based on these repurposed drugs targeting the active sites of SARS-CoV-2 proteases and for treatment options employing MG-101, Nelfinavir, Lycorine HCl, with the potential to synergistically inhibit SARS-CoV-2 replication.

## Methods

**Cell lines and virus.** HEK293T (human embryonic kidney) cells and Vero E6 (African green monkey kidney) cells were obtained from American Type Culture Collection (ATCC). Huh-7.5 (human hepatoma) cells were obtained from Dr. Charles Rice. Cells were maintained at 37 °C and 5% $CO_2$ in Dulbecco's modified Eagle's medium (DMEM, Gibco, #12800-082) supplemented with 10% fetal bovine serum (FBS, Seradigm #1500-500) and nonessential amino acids (Gibco, #11140-050). The USA-Washington strain of SARS-CoV-2 (SARS-CoV-2 Isolate USA-WA1/2020, NR-52281) was obtained from BEI Resources. The VOC Delta G/478K.V1 (B.1.617.2+AY.1+AY.2) of SARS-CoV-2 was obtained from Prof. Andrew Pekosz, Johns Hopkins University, Baltimore. All work with the SARS-CoV-2 has been conducted in Biosafety Level-3 conditions at the Eva J Pell Laboratory, The Pennsylvania State University, following the guidelines approved by the Institutional Biosafety Committees. Virus socks were generated using Vero E6 cells and the virus titers were determined using plaque assays. Aliquots of virus stocks were stored at −80 °C until use. Growth kinetic analyses of SARS CoV-2

USA-WA strain were determined using plaque assays. Vero E6 and Huh-7.5 cells were infected with the SARS-CoV-2 at 1 and 0.1 MOI. Virus supernatants were collected at 24, 48, and 72 h post infection (h.p.i.), and virus titers were determined by plaque assays on Vero E6 cells. Plaques were stained using crystal violet and pfu/ml were determined.

**ICP assay.** The mammalian expression plasmid DNA constructs, one each for testing the activity of SARS-CoV-2 Mpro and PLpro were synthesized from Twist Bioscience and cloned under a CMV promoter (Fig. 1a, b). To generate the inactive protease expression construct as assay controls, C145A mutation on Mpro and C1651A mutation on PLpro were introduced by Site-Directed Mutagenesis using primers (Table S2) and Phusion DNA Polymerase (NEB M0530). Mutations were confirmed by DNA sequencing. The constructs were transformed into *E. coli* NEB stable cells (NEB C3040) and plasmid stocks were prepared using Midiprep kit (Qiagen 12643) and stored in −20 °C. For ICP assay, HEK293T cells were grown in 96-well plates to 50–70% confluency. Media over cells were replaced with Opti-MEM (Gibco 22600050) supplemented with 1% FBS, 0.1% DMSO, and the drugs and cells were incubated at 37 °C. Untreated wells were included as controls. After 24 h of incubation, cells were transfected with the ICP constructs using PEI Max transfection reagent (Polysciences 24765). All experiments were performed in triplicate. In-cell protease assay plasmids were co-transfected with the ER marker mCherry Sec61 β C1 (Addgene Plasmid # 90994) to determine the colocalization of uncleaved protein with the ER[72].

**Immunofluorescence assay.** Huh 7.5 cells were plated on glass coverslips and treated with inhibitors 24 h before infection. Virus-infected cells were fixed at 24 h.p.i. using 3.7% paraformaldehyde for 15 min at room temperature and permeabilized using 0.1% Triton X-100 in phosphate-buffered saline (PBS) for 5 min. Cells were washed 3 times with 1× PBS. The primary antibody was SARS-CoV-2-specific rabbit polyclonal anti-N antibody (Genetex 135384). The secondary antibodies used were fluorescein isothiocyanate (FITC)-conjugated goat anti-rabbit antibody (Fisher 31635) in PBS with 10 mg/ml bovine serum albumin. Nuclei were stained using Hoechst stain (Pierce 62249) according to the manufacturer's instructions. Images were acquired using a Nikon A1R confocal microscope with 60× oil objective and 1.4 numerical aperture (NA) or 10× air objective. Images were processed using the NIS Elements software (Nikon), and the brightness and contrast were adjusted using nonlinear lookup tables.

**Live-cell imaging.** HEK293T cells were seeded onto a chambered coverslip with 8 wells (Ibidi) and transfected with mammalian expression plasmids of Mpro and PLpro (Fig. 1a, b). Cells were imaged after media were replaced with Opti-MEM reduced-serum medium (Invitrogen). Live-imaging-compatible Hoechst stain was used to stain nuclei[73]. Live imaging was conducted using a heated 60× oil immersion objective (1.4 NA) in a live imaging chamber (Tokai Hit, Fujinomiya, Shizuoka Prefecture, Japan) supplied with 5% $CO_2$ at 37 °C. The lasers and emission band-passes used for imaging were as follows: blue, excitation of 405 nm and emission of 425–475 nm; green, excitation of 488 nm and emission of 500–550 nm; red, excitation of 561 nm and emission of 570–620 nm. NIS-Elements software was used for image acquisition and analysis. Images were quantified using ImageJ (NIH, Bethesda, MD, USA).

**Antiviral inhibition assay.** Huh-7.5 cells were grown in DMEM supplemented with 10% FBS and non-essential amino acids at 37 °C and 5% $CO_2$. The cells were pre-treated with compounds at different concentrations (0.001, 1, 10, or 20 μM) or a combination of compounds for 24 h and then infected with SARS-CoV-2 at an MOI 0.1 for 1 h at 37 °C. Following infection, virus-containing media were replaced with fresh OptiMEM growth medium supplemented with inhibitors (Table S1). For the time of the additional experiment, cells were pre-treated with compounds 24 h before viral attachment, and the virus and drug-containing media was replaced with fresh media containing inhibitors and incubated for 24 h. In no-treatment experiments, cells were maintained in a compound-free medium before virus infection. After virus infection, the media were replaced with fresh media containing the compounds and incubated for another 24 h. For all experiments, cells were infected with SARS-CoV-2 at an MOI of 0.1, and virus yield in the infected cell supernatants collected at 24 h.p.i. was quantified by standard plaque assays on Vero E6 monolayers and counting the plaques after crystal violet staining.

**Cytotoxicity assay.** Approximately 10,000 cells of HEK293T or Huh-7.5 cells were plated on each well of a 96 well plate in OptimMEM + 10% FBS and incubated for 24 h at 37 °C and 5% $CO_2$. Media over cells were replaced with new media containing drugs at different concentrations (50 μl/well) and the plates were incubated for 24 h at 37 °C and 5% $CO_2$. Media without drugs were included as a negative control. All treatments were made in triplicate. After incubation, 50 μl of a 1:4 vol/vol mixture of OptiMEM and alamarBlue cell Viability Reagent (Thermo Scientific #88952) was added to each well and incubated for 4 h at 37 °C and 5% $CO_2$. Fluorescence of alamarBlue dye was quantified by measuring fluorescence Em590 at Ex545 in a Spectramax M5 plate reader. Percentage reduction was calculated as (Em590 of test − Em590 of untreated control)/(Em590 of completely reduced dye-Em590 of untreated control) × 100.

**Plaque assay.** Approximately $3 \times 10^5$ Vero E6 cells were seeded in 24-well plates and grown to form a confluent monolayer. Cells were infected with serial dilutions of virus stocks in Minimum Essential Media (MEM, GIBCO, 41500-018) supplemented with 2% FBS was added to Vero E6 monolayers on 24-well plates (Greiner bio-one, 662160) and rocked for 1 h at room temperature. The cells were then overlaid with MEM containing 1% cellulose (Millipore Sigma, 435244), 2% FBS, and 10 mM HEPES buffer, pH 7.5 (Sigma H0887), and the plates were incubated at 37 °C for 48 h. The cells were fixed with a mixture of 5% formaldehyde and 1% methanol (v/v in water) for 2 h, washed once with PBS, and stained with 0.1% Crystal Violet (Millipore Sigma V5265) prepared in 20% ethanol. After 15 min, the wells were washed with PBS, and plaques were counted to determine virus titers.

**Quantitative RT-PCR (qRT-PCR).** Effect of compounds on inhibiting SARS-CoV-2 replication in Huh-7.5 cells via inhibition of Mpro and PLpro was tested by comparing the levels of viral RNA in treated and untreated cells at 24 h.p.i. Huh-7.5 cells grown in DMEM in 6 well tissue culture plates were pre-treated with different concentrations of compounds for 24 h were infected with 0.1 MOI of SARS-CoV-2. After 1 h of infection, virus-containing media were replaced with fresh media-containing compounds, and the plates were incubated at 37 °C. Untreated cells were included in the experiment as controls. After 24 h, total RNA was extracted from the cells using a Zymopure Quick-RNA extraction kit. The number of viral RNA molecules in the extracted total RNA samples was determined by quantitative RT-PCR on a QuantStudio-3 machine (Applied BioSystems) using primers –Forward GTGAAATGGTCATGTGTGGCGG and Reverse CAAATGT-TAAAAACACTATTAGCATA. The Ct values obtained were used to calculate the number of RNA molecules using a standard curve of Ct versus RNA molecules.

**Cloning, expression, and purification of SARS-CoV-2 Mpro.** Gene encoding the full-length SARS-CoV-2 Mpro (residues 1–306) was cloned into pSUMO vector (Life Sensors) using *BsaI* and *XhoI* restriction sites to make His$_6$-SUMO-Ulp1 recognition site- Mpro fusion. A 1 L culture containing BL21(DE3) cells transformed with this construct was grown at 37 °C in LB medium supplemented with 50 μg/mL Kanamycin. The protein expression was induced by adding 0.5 mM IPTG at OD$_{600}$ = 0.6, and the culture was incubated overnight at 18 °C. Cells were harvested by centrifugation and resuspended in 30 mL of Buffer A [20 mM Tris-HCl (pH 8.0 at 4 °C), 500 mM NaCl, 5% Glycerol, 0.5 mM DTT] and lysed by sonication. The lysate was clarified by centrifugation (146k×g) at 4 °C for 1 h. The supernatant was loaded on HisTrap FF column (GE Healthcare) equilibrated with buffer A followed by washing with buffer A. The fusion protein was eluted with buffer B (20 mM Tris-HCl, 200 mM NaCl, 300 mM imidazole, 0.5 mM DTT). Fractions containing His$_6$-SUMO-Mpro protein were incubated with Ulp1 protease at a molar ration 20:1 for 2 h at 30 °C. The cleaved protein was dialyzed with buffer C (20 mM Tris, 100 mM NaCl, 5% Glycerol 0.5 mM DTT) followed by passing through HisTrap FF column to remove the His$_6$-SUMO tag and uncleaved fusion protein. The Mpro contains Ser residue at its N-terminus as the natural Mpro processed from the SARS-CoV-2 polyproteins. Mpro is further purified by Superdex200 (26/600) size exclusion column (GE Healthcare) chromatography in activity buffer [20 mM HEPES (pH 7.5), 100 mM NaCl, 5% Glycerol, 1 mM DTT].

**Cloning, expression, and purification of SARS-CoV-2 PLpro protease.** Gene encoding the full-length SARS-CoV-2 PLpro (residues 746-1064) was cloned into pSUMO vector (Life Sensors) using BsaI and XhoI restriction sites to make His$_6$-SUMO-Ulp1 recognition site-Mpro fusion. BL21(DE3) cells transformed with pSUMO-PLpro plasmid were grown at 37 °C overnight in 5 mL LB medium supplemented with 50 μg/mL Kanamycin. 1 L LB broth media was inoculated with the overnight grown cells and incubated at 37 °C with shaking. The protein expression was induced by adding 0.5 mM IPTG at OD$_{600}$ = 0.6, and the culture was incubated overnight at 18 °C. Cells were harvested by centrifugation and resuspended in 30 mL of Buffer A (20 mM Tris-HCl (pH 8.0 at 4 °C), 500 mM NaCl, 5% Glycerol, 5 mM DTT, 1 μM ZnCl$_2$) and lysed by sonication. The lysate was clarified by centrifugation (146k×g) at 4 °C for 1 h. The supernatant was then loaded on HisTrap FF column (GE Healthcare) equilibrated with buffer A. followed by washing with buffer A, the fusion protein was eluted with buffer B (20 mM Tris-HCl, 200 mM NaCl, 300 mM imidazole, 5 mM DTT, 1 μM ZnCl$_2$). Fractions containing His$_6$-SUMO-PLpro protein were incubated with Ulp1 protease at a molar ration 20:1 for 2 h at 30 °C. The cleaved protein was dialyzed with buffer C (20 mM Tris, 100 mM NaCl, 5% Glycerol, 5 mM DTT, 1 μM ZnCl$_2$) followed by passing through HisTrap FF column to remove the His$_6$-SUMO tag and uncleaved fusion protein. The PLpro protease is further purified by Superdex200 (26/600) size exclusion column (GE Healthcare) chromatography in activity buffer (20 mM HEPES (pH 7.5), 100 mM NaCl, 1 mM DTT).

**In vitro Mpro protease inhibition assay.** For the determination of half-maximal inhibitory concentrations (IC$_{50}$) of MG-101 and GC376 (BPS Bioscience), a synthetic substrate of Mpro containing FRET pair and Mpro nsp4/5 cleavage site (indicated by an arrow, Dabcyl-KTSAVLQ ↓ SGFRKME-Edans) (BPS Bioscience) was used for the FRET-based cleavage assay[74]. The assay was performed in 96-well, black, flat-bottomed microtiter plates (Greiner Bio-One, Germany) with a final volume of 50 μl. Mpro (a final concentration of 200 nM) was pre-incubated for 1 h

at 22 °C with compounds at different concentrations in the assay buffer (20 mM HEPES (pH 7), 0.5 mM EDTA, 1 mM dithiothreitol, 5% glycerol). The substrate was then added at a final concentration of 10 μM to the reaction mixture and the reaction was incubated for 4 h at 22 °C. The readings for the different concentrations of the inhibitor compounds incubated with the substrate without Mpro were measured as a blank. The fluorescence signals (excitation/emission, 360 nm/ 460 nm) of released EDANS were measured using Infinite 200 PRO multimode plate reader (Tecan, USA). The results were plotted as dose inhibition curves using nonlinear regression with a variable slope to determine the IC$_{50}$ values of inhibitor compounds using GraphPad Prism 9.0.

**In vitro PLpro protease inhibition assay.** For the determination of half-maximal inhibitory concentrations (IC$_{50}$) of inhibitors against the PLpro protease, a peptide-AMC substrate based on the C-terminal residues of ubiquitin Z-RLRGG-AMC (BPS Bioscience) was used for the fluorescence-based cleavage assay (58). The assay was performed in 96-well, black, flat-bottomed microtiter plates (Greiner Bio-One, Germany) with a final volume of 50 μl. PLpro (a final concentration of 300 nM) was pre-incubated for 1 h at 25 °C with compounds at different concentrations in the assay buffer (20 mM HEPES (pH 7.5), 1 mM DTT, 100 mM NaCl). The substrate was then added at a final concentration of 25 μM to the reaction mixture and the reaction was incubated for 1 h at 25 °C. The readings for the different concentrations of the inhibitor compounds incubated with the substrate without PLpro were measured as a blank. The fluorescence signals (excitation/emission, 340 nm/ 460 nm) of released AMC were measured using Infinite 200 PRO multimode plate reader (Tecan, USA). The results were plotted as dose inhibition curves using nonlinear regression with a variable slope to determine the IC$_{50}$ values of inhibitor compounds using GraphPad Prism 9.0.

**Crystallization of Mpro and Mpro-MG-101 complex.** Mpro (25 mg/ml) in buffer C [20 mM HEPES (pH 7.5), 5% Glycerol, 100 mM NaCl, 0.5 mM DTT] was used for screening crystallization conditions by using sitting drop vapor diffusion. By mixing equal volume of the Mpro solution and PACT premier crystal screen solutions (Molecular Dimensions), crystals formed after four days in E8 (0.2 M sodium sulfate, 20% PEG 3350), E9 (0.2 M potassium sodium tartrate, 20% PEG 3350) and E11 (0.2 M sodium citrate, 20% PEG 3350) conditions and these crystals were used as seeds to grow better quality crystals. Crystals used for X-ray data collection were grown by using a solution containing 0.2 M sodium sulfate, 10% PEG 3350. Two different crystal forms (C2 and P21) were obtained in these crystallization droplets. The crystals were soaked in a cryoprotectant solution consisting of 0.2 M sodium sulfate/citrate, 10% PEG 3350, 20 mM HEPES (pH 7.5), and 15% glycerol for 10 min, followed by flash freezing in the liquid nitrogen. To crystallize the Mpro and MG-101 complex, Mpro (25 mg/ml) in buffer C was incubated with 7 times molar concentration of the inhibitor for 4 days at 4 °C. The mixture was then centrifuged at 13,000 g for 10 min to remove any aggregates, followed by screening crystallization condition by using sitting-drop vapor diffusion and commercial crystallization screening solutions. PACT premier screen (Molecular Dimensions) conditions E8 (0.2 M sodium sulfate, 20% PEG 3350), E9 (0.2 M potassium sodium tartrate, 20% PEG 3350) and E11 (0.2 M sodium citrate, 20% PEG 3350) as well as Crystal screen I and II (Hampton Research) conditions B3 (0.2 M ammonium sulfate, 0.1 M sodium cacodylate trihydrate pH 6.5, 20% PEG 8000) and G2 (0.2 M ammonium sulfate, 0.1 M MES monohydrate pH 6.5, 30% polyethylene glycol monomethyl ether 5000) formed crystals. Rectangular and multi-faceted crystals of 0.1–0.2 mm size grew in 2 days. Crystals were soaked in cryoprotectant solution consisting of 0.2 M sodium sulfate, 15% PEG-3350, 5–15% DMSO, 20 mM HEPES (pH 7.5) for 10 min then flash-frozen by liquid nitrogen.

**X-ray data collection and structure determination.** X-ray diffraction data were collected at the Macromolecular Diffraction at Cornell High Energy Synchrotron Source (MacCHESS) ID7A1 beamline (Cornell University, Ithaca, NY), and the data were processed by HKL2000[75]. The structure of SARS-CoV-2 Mpro (PDB 6WTK) was used as a search model for the molecular replacement using Phaser in Phenix[29,76]. The structures were refined by using Phenix for the rigid body and positional refinements with reference structure restraints to avoid over-fitting the data[77]. Final coordinates and structure factors were submitted to the PDB depository with ID codes listed in Table 1. Figures reporting structures of Mpro were prepared using Chimera [78].

**Docking method.** The X-ray crystal structures of Mpro (7BQY) and PLpro (6WX4) in complex with potent covalent peptidomimetic inhibitors were downloaded from the Protein Data Bank. Both proteins were prepared using the Protein Preparation Wizard in Maestro (Schrödinger Release 2019-3, https://www.Schrodinger.Com), waters, and other co-crystallized molecules were removed, except for the ligand. Predicting protonation states of protein residues were calculated, considering a temperature of 300 °K and a pH of 7. The ligands were prepared using the Ligprep tool considering the ionization states at pH 7 ± 2. A 15 Å docking grid (inner-box 10 Å and outer-box 20 Å) was prepared using as centroid the co-crystallized ligand. The docking studies were performed using Glide SP precision, keeping the default parameters and setting, and it was combined with "molecular mechanics generalized Born surface area" (MMGBSA),

implemented in the Prime module from Maestro to re-score the three-output docking poses of each compound. In the case of compound MG-101, a further analysis was performed using Covalent docking simulations implemented in Maestro (Covalent Dock Lead Optimization workflow) and the best poses were rescored with MMGBSA. Molecular Operating Environment 2019.1 (MOE) was used to visualize the structures and acquire the images.

**Statistics and reproducibility**. All statistical analyses were performed using GraphPad Prism 9.0. Plaque assays and cytotoxicity assays were performed in triplicates of one biological experiment. The $EC_{50}$ values of inhibitors were calculated from dose inhibition curves using nonlinear regression with a variable slope using GraphPad Prism. The number of viral RNA molecules in samples was calculated by qRT-PCR experiments using three samples, and data were analyzed by GraphPad Prism 9.0. The $IC_{50}$ values of inhibitor compounds of Mpro and PLpro proteases were calculated using GraphPad Prism 9.0. from dose inhibition curves using nonlinear regression with a variable slope.

**Reporting summary**. Further information on research design is available in the Nature Research Reporting Summary linked to this article.

## Data availability
The atomic coordinates and structure factors have been deposited in the Protein Data Bank with ID codes 7LKE, 7LKD, and 7LBN. Computational data are available on request from the corresponding author. All other data supporting the findings of this study are available within the paper and Supplementary Data 1. Any remaining data can be obtained from the corresponding author upon reasonable request.

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

## Acknowledgements
We thank the staff at the MacCHESS for support of crystallographic data collection. We thank Dr. Andrew Pekosz for providing us with the SARS-CoV-2 variants, Dr. Miyashiro for using the Infinite 200 PRO multimode plate reader, and Drs. Girish S. Kirimanjeswara and M. Joanne Lemieux for scientific discussions. This work was supported by NIH grant R35 GM131860 to K.S.M, Welsh Government Office for Science Sêr Cymru Tackling COVID-19 grant for CV, COVID-19 seed funding from the Huck Institutes of the Life Sciences and Penn State start-up funds to J.J.

## Author contributions
A.N. carried out molecular lab work, imaging, antiviral studies in BSL-3, data analysis, and drafted the paper. M.N. cloned and purified Mpro and PLpro, performed biochemical assays, crystallization, structure determination, and helped draft the paper. S.A.M. performed BSL-3 experiments and molecular lab work. C.V. performed docking experiments and participated in drafting the paper. S.A.T. participated in BSL-2 inhibitor studies. C.B. participated in experimental design, data analysis, and paper drafting. A.B. conceived and coordinated the docking studies and participated in drafting the paper. K.S.M. designed the structural studies, participated in crystallization and structure determination, and drafted the paper. J.J. conceived and coordinated the study, participated in antiviral studies, data analysis, and drafted the manuscript. All authors discussed the experiments, results and approved the paper.

## Competing interests
The authors declare no competing interests.
