## [Peer Review File · Communications Biology]

Reviewers' comments:

Reviewer #1 (Remarks to the Author):

In this manuscript by Narayanan et al, the authors carried out an in-cell protease assay, cell-based antiviral assays, biochemical activity assessments, and crystallography to identify inhibitors targeting SARS-CoV-2 Mpro and PLpro. They have identified a couple of enzyme inhibitors with antiviral activity. Further the authors describe the binding of MG-101 to SARS-CoV-2 Mpro in atomic detail, respectively. Altogether, this work aims to address a very urgent public health need since those viral proteases from SARS-CoV-2 are considered to be attractive targets for drug development. This is another attempt targeting viral proteases following the previous studies (Zhang et al., Science, 2020; Jin et al., Nature, 2020; Dai et al., Science, 2020; Jin et al., NSMB, 2020; Qiao et al., Science, 2021; Zhao et al., Protein Cell, 2021; Osipiuk et al., Nat Commun, 2021; Huang et al., Nat Commun, 2021; Shin et al. Nature, 2020; Klemm et al., EMBO J, 2020, Rut et al., Sci Adv, 2020). The work is interesting, but there are some major concerns which need to be addressed.

1. The crystal structure has unambiguously show that MG-101 forms covalent bond with C145, indicating that this compound belongs to time-dependent inhibitors. The authors need to measure both the equilibrium-binding constant K_i and the inactivation rate constant k_{inac} for covalent bond formation to evaluate this type of inhibitors.
2. Please provide omit map for the inhibitor in the supplemental materials.
3. Please provide images for the plaque-reduction assays in the supplemental materials.
4. Biochemical activity assays are needed to confirm those inhibitors from initial screening are targeting SARS-CoV-2 PLpro.
5. The validation report for the crystal complex structure should be uploaded.

Reviewer #2 (Remarks to the Author):

In this manuscript, Narayanan et al. used in-cell protease (ICP) assays to test the inhibition of 64 FDA-approved pharmaceutical drugs in SARS-CoV-2 Mpro and PLpro viral protease activity in HEK293T cell culture.

The authors constructed a fluorescent protein with nuclear localization signal (mEmerald-NLS) linked to an ER membrane anchoring protein (ZIKV NS2B) and Mpro and PLpro proteases (wt and dead mutants). Upon protease cleavage in trans, mEmerald-NLS is released from the construct and localizes from ER membrane to the nucleus. Reduction of nuclear localization of mEmerald-NLS due to viral protease inhibition was quantified by live-cell confocal microscopy. The authors also evaluated the cytotoxicity of the compounds with cell viability assays using alamarBlue. With selected targets, the authors tested the inhibition effects of the compounds on SARS-CoV-2 replication in Huh 7.5 cells by plaque assays, immunofluorescence assays and qRT-PCR. To further elucidate the interaction between the selected inhibitors and the viral proteases, the authors also used molecular docking analyses and X-ray crystallography for one compound, and examined in-vitro inhibition of the proteases by performing FRET assays. In the end, the authors identified 7 protease inhibitors from the list of 64 pre-existing drugs and modeled their interaction with the appropriate protease active site. For the papain-like protease (PLpro) these inhibitors are Sitagliptin and Daclatasvir, and the main protease (Mpro) they are MG-101, Lycorine HCl and Nelfinavir mesylate.

At first glance, the design of the study and the results presented warrants publication. However, the ambiguous description about the novelty of the study is a concern, considering all the data that can be found in the literature about SARS-CoV-2 protease inhibitors. The authors did a very poor job describing what has been done before, and what are the gap they are filling. Specifically, the main compound presented here, MG-101, is also called Calpain protease inhibitor I in previous SARS-CoV-2/Mpro paper (Ma et al. Cell Research 2020, Sacco et al. Science advances 2020), as well as ALLN (Daniloski et al., Cell 2020), another SARS-CoV-2 paper. The way this work is presented makes it very hard to realize that this compound presented here is not new, with FRET assays and infectivity data already published. Perhaps the novelty is the crystal the structure;

which is fine, but the authors need to re-write the entire manuscript to present their results considering previous studies.

Comments:

- Line 124. "The Zika virus NS2B protein anchors the uncleaved fluorescent protein to the ER membrane."

 Based on Figure 1, it really doesn't look like the construct localizes to the ER. The authors should either show the ER co-localization, or acknowledge that their construct didn't work as expected. I don't think it's necessarily a deal breaker, as they are measuring the translocation from what it looks like the cytoplasm to the nucleus.

- How many compounds, out of 64, have been tested before to inhibit the Mpro in vitro or in cell? how about PLpro? Or tested against SARS-CoV-2? The authors must present a better picture about previous studies.

- Why use Huh-7.5 cells? Why not (also) use A549 or another alveolar cell line for the respiratory virus SARS-CoV-2? The authors should comment.

- Page 8, line 187: Although authors investigated the additive effects of 3 Mpro inhibitors on virus titres in plaque assays, the authors did not perform further experiments or discuss the potential of combining both Mpro and PLpro inhibitors as a treatment. Have you also examined the cytotoxicity effects when combining 2 of the 3 inhibitors MG-101, Nelfinavir mesylate and Lycorine HCl?

- On page 11, line 291 "FRET assay using Dabcyl-Edans fluorescence pair was performed to validate potential Mpro inhibitors showing antiviral activity through the in-cell protease assay."

 Why not performing similar assay for PLpro inhibitors identified by the in-cell protease assay, as well as for the other Mpro inhibitors identified?

- I would recommend the authors present the crystal structure first, and then present the docking models for the complexes without structures.

- Line 488. "MG-101 is a potent inhibitor of cysteine proteases, which inhibits calpain I, calpain II, cathepsin B and cathepsin L. Our results indicate that MG-101 inhibits SARS-CoV-2 replication by inhibiting Mpro activity with an EC50 of 0.038 μ M. A previous study showed that Calpain inhibitors II and XII inhibit SARS-CoV-2 in the CPE assay with EC50 values of 2.07 and 0.49 μ M, respectively by inhibiting Mpro activity (41)."

 And this same study also showed that compound 59, which is calpain inhibitor I (or MG-101) also inhibit Mpro at 8.6 μ M

Minor comments:

- Line 41 "Currently, the antiviral strategies against SARS coronaviruses are mainly focused on small molecules to prevent spike-receptor binding, interfere with RNA-dependent RNA polymerase (RdRP) activity, and inhibit the two viral proteases (9). Even though all viral enzymes that participate in coronavirus replication are potentially druggable targets, antiviral studies with small-molecule inhibitors have been focused on RdRP and the two viral proteases PLpro and Mpro (10) (11, 12)."

 These two sentences are redundant.

Response to the reviewer

We have completed the revision of our manuscript ‘Identification of SARS-CoV-2 inhibitors targeting Mpro and PLpro using in-cell-protease assay’. We thank the reviewers for the constructive comments on our previous version of the manuscript. We have restructured our manuscript addressing technical concerns, performed additional experiments, and provided new data in the revised manuscript. We believe that addressing the concerns of the reviewers has significantly improved the quality and strength of the manuscript.

Please find below our point-by-point responses to the reviewers’ comments. The italicized blue font represents the response.

Reviewer #1 (Remarks to the Author):

In this manuscript by Narayanan et al, the authors carried out an in-cell protease assay, cell-based antiviral assays, biochemical activity assessments, and crystallography to identify inhibitors targeting SARS-CoV-2 Mpro and PLpro. They have identified a couple of enzyme inhibitors with antiviral activity. Further the authors describe the binding of MG-101 to SARS-CoV-2 Mpro in atomic detail, respectively. Altogether, this work aims to address a very urgent public health need since those viral proteases from SARS-CoV-2 are considered to be attractive targets for drug development. This is another attempt targeting viral proteases following the previous studies (Zhang et al., Science, 2020; Jin et al., Nature, 2020; Dai et al., Science, 2020; Jin et al., NSMB, 2020; Qiao et al., Science, 2021; Zhao et al., Protein Cell, 2021; Osipiuk et al., Nat Commun, 2021; Huang et al., Nat Commun, 2021; Shin et al. Nature, 2020; Klemm et al., EMBO J, 2020, Rut et al., Sci Adv, 2020). The work is interesting, but there are some major concerns which need to be addressed.

We have incorporated changes in the revised manuscript and have provided answers to the critiques in italics. We thank the reviewer for recognizing the significance of our findings as antivirals are urgently required for new drug development for combating ongoing pandemic and any future SARS-CoV-2 outbreaks. In the revised manuscript, we have added relevant new studies in antivirals development targeting the SARS-CoV-2 proteases and incorporated the references specifically mentioned by the reviewer #1.

1. The crystal structure has unambiguously show that MG-101 forms covalent bond with C145, indicating that this compound belongs to time-dependent inhibitors. The authors need to measure both the equilibrium-binding constant K_i and the inactivation rate constant k_{inac} for covalent bond formation to evaluate this type of inhibitors.

We are thankful to this reviewer for suggesting the determination of K_i and k_{inac} for the MG101 binding to the Mpro protease. Since a goal of this study is to screen and identify the inhibitors against SARS CoV-2 proteases, in line with other similar published studies (1), we did not address the kinetics for the inhibitors against proteases identified in this study. The IC_{50} values

provide a convenient and simple means to interpret the inhibition by the screened compounds in this research. We thus did not evaluate the K_i and k_{inac} measurements for the compounds.

(1. Zhang, L., Lin, D., Sun, X., Curth, U., Drosten, C., Sauerhering, L., Becker, S., Rox, K. and Hilgenfeld, R., 2020. Crystal structure of SARS-CoV-2 main protease provides a basis for design of improved α -ketoamide inhibitors. Science, 368(6489), pp.409-412.)

2. Please provide omit map for the inhibitor in the supplemental materials.

The omit map of MG-101 bound at the active site Mpro is added in the supplementary figures (Figure S3)

3. Please provide images for the plaque-reduction assays in the supplemental materials.

We thank the reviewer for this thoughtful comment. We are measuring the reduction in virus titer by plaque assay. Therefore, plaque sizes remain unchanged while the number of plaques is reduced from the untreated cells. The virus titers represented in Fig 4 indicate the actual reduction in virus titer. Therefore, we do not find it is necessary to add several plaque assay images of all inhibitors. Although we have recorded images of all plaque assays we performed in BSL-3, we are limited by our ability to obtain high-resolution images of plaque assays from the BSL-3 laboratory. We include a figure for the reviewer to prove our ability to perform these experiments and obtain reliable data. However, we can include additional images in the manuscript if deemed necessary.

Figure 1. Representative image of a plaque assay to determine SARS-CoV-2 titer.

Cell-culture supernatants from Huh7.5 cells treated with drugs and infected with SARS-CoV-2 were collected 24 h.p.i . Plaque assays were performed as described in materials and methods. Numbers 4, 6, 7, 23, 25, and 29 labeled on the left-hand side are codes given by the researcher for drug treatments. 1-4 labeled on top are dilutions, and the plate is labeled 10B by the researcher.

4. Biochemical activity assays are needed to confirm those inhibitors from initial screening are targeting SARS-CoV-2 PLpro.

We appreciate a suggestion by this reviewer for performing the activity assay in vitro to confirm the inhibition of the proteolytic activity of the PLpro. Fluorogenic substrate peptide Z-Arg-Leu-Arg-Gly-Gly-AMC, was used for testing the effect on the proteolytic activity of the PLpro by the screened inhibitors. Appropriately the data has been included in the manuscript.

5. The validation report for the crystal complex structure should be uploaded.

The validation reports of the crystal structures are attached for the reference.

Reviewer #2 (Remarks to the Author):

In this manuscript, Narayanan et al. used in-cell protease (ICP) assays to test the inhibition of 64 FDA-approved pharmaceutical drugs in SARS-CoV-2 Mpro and PLpro viral protease activity in HEK293T cell culture.

The authors constructed a fluorescent protein with nuclear localization signal (mEmerald-NLS) linked to an ER membrane anchoring protein (ZIKV NS2B) and Mpro and PLpro proteases (wt and dead mutants). Upon protease cleavage in trans, mEmerald-NLS is released from the construct and localizes from ER membrane to the nucleus. Reduction of nuclear localization of mEmerald-NLS due to viral protease inhibition was quantified by live-cell confocal microscopy. The authors also evaluated the cytotoxicity of the compounds with cell viability assays using alamarBlue. With selected targets, the authors tested the inhibition effects of the compounds on SARS-CoV-2 replication in Huh 7.5 cells by plaque assays, immunofluorescence assays and qRT-PCR. To further elucidate the interaction between the selected inhibitors and the viral proteases, the authors also used molecular docking analyses and X-ray crystallography for one compound, and examined in-vitro inhibition of the proteases by performing FRET assays. In the end, the authors identified 7 protease inhibitors from the list of 64 pre-existing drugs and modeled their interaction with the appropriate protease active site. For the papain-like protease (PLpro) these inhibitors are Sitagliptin and Daclatasvir, and the main protease (Mpro) they are MG-101, Lycorine HCl and Nelfinavir mesylate.

At first glance, the design of the study and the results presented warrants publication.

We thank the reviewer for acknowledging that our study warrants publication with suggested changes.

However, the ambiguous description about the novelty of the study is a concern, considering all the data that can be found in the literature about SARS-CoV-2 protease inhibitors.

We thank the reviewer for pointing out the shortcomings in describing the novelty of our study. The novelty of our study is in developing the ICP assay for screening the inhibitors, and subsequent identification of drugs against Mpro and PLpro. In this revised manuscript, we have also included new data on the effects of the selected inhibitors and inhibitor combinations on SARS-CoV-2 delta variant. We have made changes in the text to reinforce the novelty of this study.

The authors did a very poor job describing what has been done before, and what are the gap they are filling. Specifically, the main compound presented here, MG-101, is also called Calpain protease inhibitor I in previous SARS-CoV-2/Mpro paper (Ma et al. Cell Research 2020, Sacco et al. Science advances 2020), as well as ALLN (Daniloski et al., Cell 2020), another SARS-CoV-2 paper. The way this work is presented makes it very hard to realize that this compound presented here is not new, with FRET assays and infectivity data already published. Perhaps the novelty is the crystal the structure; which is fine, but the authors need to re-write the entire manuscript to present their results considering previous studies.

We thank the reviewer for pointing out our lack of clarity in describing previous work. We have thoroughly revised the manuscript indicating prior work and describing MG-101 as Calpain inhibitor I and referenced the publications mentioned by the reviewer.

The novelty of this manuscript is also in the identification of new compounds that can be developed into more potent inhibitors. We also show that while some of the compounds are not new, their superior effect in combined inhibition in reducing the virus titer is better than the existing data. Also, when used in combination inhibition, we show that they have a significant antiviral effect against the SARS-CoV-2 delta variant. We have highlighted these critical findings in the revised manuscript.

Comments:

- Line 124. “The Zika virus NS2B protein anchors the uncleaved fluorescent protein to the ER membrane.”

 Based on Figure 1, it really doesn't look like the construct localize to the ER. The authors should either show the ER co-localization, or acknowledge that their construct didn't work as expected. I don't think it's necessarily a deal breaker, as they are measuring the translocation from what it looks like the cytoplasm to the nucleus.

We thank the reviewer for the suggestion. We have performed new imaging experiments to show the co-localization of proteins with ER marker. We have revised Figure 1 describing the ICP assay, which now shows the co-localization of the uncleaved mEmerald-NLS protein (green) with the endoplasmic reticulum marker mCherry-Sec61β (red).

How many compounds, out of 64, have been tested before to inhibit the Mpro in vitro or in cell? how about PLpro? Or tested against SARS-CoV-2? The authors must present a better picture about previous studies.

We have modified the introduction and discussion as per the reviewer's suggestion for better clarity on the numbers of compounds used in each assay. We have also modified the introduction to incorporate previous studies.

Why use Huh-7.5 cells? Why not (also) use A549 or another alveolar cell line for the respiratory virus SARS-CoV-2? The authors should comment.

We have revised the manuscript describing the use of Huh 7.5 cells in our studies. We have tested A549 cells and Calu-3 cells in our assays; however, the cells were not consistently producing high titer virus, giving false-positive results. We needed consistent and robust virus production from a human cell line; therefore, we chose Huh 75 cells for our studies. Additionally, as we mentioned in the manuscript, drugs that were initially identified using the ICP assay using HEK293T cells showed inhibitory activity against SARS-CoV-2 in all cell types we tested.

We have used several cell lines for experiments with SARS-CoV, as shown in our previously published studies (Clausen et al. 2020)

Clausen, T.M., et al., SARS-CoV-2 Infection Depends on Cellular Heparan Sulfate and ACE2. Cell, 2020. 183(4): p. 1043-1057 e15.

Page 8, line 187: Although authors investigated the additive effects of 3 Mpro inhibitors on virus titres in plaque assays, the authors did not perform further experiments or discuss the potential of combining both Mpro and PLpro inhibitors as a treatment. Have you also examined the cytotoxicity effects when combining 2 of the 3 inhibitors MG-101, Nelfinavir mesylate and Lycorine HCl?

We thank the reviewer for this helpful comment. We have performed new experiments, collected data, and provided new figures to show this (Figure 6 and Figure S2). We have examined the cytotoxicity of every drug combination we have used in our experiments, as shown in Fig S2C.

On page 11, line 291 "FRET assay using Dabcyl-Edans fluorescence pair was performed to validate potential Mpro inhibitors showing antiviral activity through the in-cell protease assay."  Why not performing similar assay for PLpro inhibitors identified by the in-cell protease assay, as well as for the other Mpro inhibitors identified ?

We appreciate a suggestion by this reviewer for performing the activity assay in vitro to confirm the inhibition of the proteolytic activity of the PLpro. Fluorogenic substrate peptide Z-Arg-Leu-Arg-Gly-Gly-AMC, was used for testing the effect on the proteolytic activity of the PLpro by the screened inhibitors. Appropriately the data has been included in the manuscript.

I would recommend the authors present the crystal structure first, and then present the docking models for the complexes without structures.

We appreciate the comment of this reviewer. However, docking experiments were performed using previously reported crystal structures. The crystal structure of MG-101 bound to Mpro we report here was to understand the mechanism of inhibition of MG-101. We believe that the current format is more helpful in explaining the mechanism of action of the compounds and the combined inhibition of various inhibitors used in the study. Therefore we have not changed the order of results in the revised manuscript.

Line 488. “MG-101 is a potent inhibitor of cysteine proteases, which inhibits calpain I, calpain II, cathepsin B and cathepsin L. Our results indicate that MG-101 inhibits SARS-CoV-2 replication by inhibiting Mpro activity with an EC50 of 0.038 μ M. A previous study showed that Calpain inhibitors II and XII inhibit SARS-CoV-2 in the CPE assay with EC50 values of 2.07 and 0.49 μ M, respectively by inhibiting Mpro activity (41).”

 And this same study also showed that compound 59, which is calpain inhibitor I (or MG-101) also inhibit Mpro at 8.6 μ M

We thank the reviewer for this helpful comment. We have included this report in the manuscript and highlighted the importance of this finding.

Minor comments:

Line 41 “Currently, the antiviral strategies against SARS coronaviruses are mainly focused on small molecules to prevent spike-receptor binding, interfere with RNA-dependent RNA polymerase (RdRP) activity, and inhibit the two viral proteases (9). Even though all viral enzymes that participate in coronavirus replication are potentially druggable targets, antiviral studies with small-molecule inhibitors have been focused on RdRP and the two viral proteases PLpro and Mpro (10) (11, 12).”

 These two sentences are redundant.

We have revised this section of the manuscript to avoid redundancy.

REVIEWERS' COMMENTS:

Reviewer #1 (Remarks to the Author):

The authors have addressed the concerns from the reviewer.

Reviewer #2 (Remarks to the Author):

The authors have addressed my concerns.

The authors should update their introduction regarding FDA approval drugs (e.g. paxlovid), and perhaps write it in a way that it will still make sense a few years from now considering how fast this field is changing. Albeit not necessary for publication, I wonder how their structure compares to other Mpro structure with MG- compounds (e.g. MG-132, pdb 7NG3). Perhaps the authors could comment in the discussion.

RE: Final revisions for manuscript COMMSBIO-21-1617A-Z

Note: Author's responses are in blue font and italics.

REVIEWERS' COMMENTS:

Reviewer #1 (Remarks to the Author):

The authors have addressed the concerns from the reviewer.

We thank the reviewer for their insightful comments that helped improve the manuscript.

Reviewer #2 (Remarks to the Author):

The authors have addressed my concerns.

We thank the reviewer for agreeing with our corrections and modifications of the revised manuscript.

The authors should update their introduction regarding FDA approval drugs (e.g. paxlovid), and perhaps write it in a way that it will still make sense a few years from now considering how fast this field is changing.

In the revised manuscript, we have included information regarding FDA approved drugs in the introduction.

Albeit not necessary for publication, I wonder how their structure compares to other Mpro structure with MG- compounds (e.g. MG-132, pdb 7NG3). Perhaps the authors could comment in the discussion.

We thank the reviewer for this suggestion. We have added a comment about the similarities of the compounds bound to the protease in the discussion.